# Bringing regularized optimal transport to lightspeed: a splitting method adapted for GPUs

**Jacob Lindbäck**
EECS, KTH
Stockholm, Sweden
jlindbac@kth.se

**Zesen Wang**
EECS, KTH
Stockholm, Sweden
zesen@kth.se

**Mikael Johansson**
EECS, KTH
Stockholm, Sweden
mikaelj@kth.se

## Abstract

We present an efficient algorithm for regularized optimal transport. In contrast to previous methods, we use the Douglas-Rachford splitting technique to develop an efficient solver that can handle a broad class of regularizers. The algorithm has strong global convergence guarantees, low per-iteration cost, and can exploit GPU parallelization, making it considerably faster than the state-of-the-art for many problems. We illustrate its competitiveness in several applications, including domain adaptation and learning of generative models.

## 1  Introduction

Optimal transport (OT) is an increasingly important tool for many ML problems. It has proven successful in a wide range of applications, including domain adaptation [8, 19], learning generative models [2, 17], smooth ranking and sorting schemes [12], and long-tailed recognition [29]. The versatility of OT stems from the fact that it provides a flexible framework for comparing probability distributions that incorporates the geometric structure of the underlying sample space [30]. Early on, practitioners were directed to LP solvers with poor scalability to solve OT problems, but this changed dramatically with the introduction of entropic regularization and the Sinkhorn algorithm for OT. Sinkhorn's simple parallelizable operations resolved a computational bottleneck of OT and enabled the solution of large-scale problems at "lightspeed" [11]. Despite the computational advantages of entropic regularization, many applications depend on sparse or structured transportation plans, and the bias that entropic regularization introduces can significantly impact the performance of the downstream task being solved. In such settings, other structure-promoting regularizers are typically preferred (see e.g. [10, 25, 19]). However, to the best of our knowledge, no framework exists that handles general regularizers for OT in a unifying way with a numerical efficiency that is comparable to Sinkhorn. To this end, we study OT for a broad class of regularizers and develop an algorithm with similar computational performance as Sinkhorn. Further, we benchmark our method against the state-of-the-art, showing that our algorithm achieves up to 100x speed-up for several applications.

For discrete probability distributions, solving regularized OT problems amounts to finding a solution to the optimization problem

$$\begin{array}{ll} \underset{X \in \mathbb{R}_+^{m \times n}}{\text{minimize}} & \langle C, X \rangle + h(X) \\ \text{subject to} & X\mathbf{1}_n = p, \ X^\top \mathbf{1}_m = q. \end{array} \tag{1}$$

Here $p$ and $q$ are non-negative vectors that sum to 1, $C$ is a non-negative cost matrix, and the regularizer $h$ is a function that promotes structured solutions. To solve (1) fast, it is important that the large number of decision variables and the non-negativity constraint on the transportation plan are handled in a memory-efficient way. It is also crucial to manage the non-smoothness induced by the non-negativity constraints without altering the complexity of the algorithm. It is common

37th Conference on Neural Information Processing Systems (NeurIPS 2023).

practice to solve the OT problem by considering its dual since many regularization terms give rise to dual problems with structure that can be exploited [5]. Most notably, entropic regularization, i.e. $h(X) = \epsilon \sum_{ij} X_{ij} \log X_{ij}$, where $\epsilon > 0$, enables deriving a simple alternating dual ascent scheme, which coincides with the well-known Sinkhorn-Knopp algorithm for doubly stochastic matrices [39]. An advantage of this scheme is that it is easy to parallelize and has a low per-iteration cost [11]. However, decreasing the regularization parameter $\epsilon$ will slow down the convergence of the algorithm and ultimately render the algorithm numerically unstable, which is particularly noticeable when low-precision arithmetic is used [27]. Conversely, increasing $\epsilon$ makes the transportation plan blurrier - which can be problematic in applications when sparse solutions are desired [5].

For more general regularizers, it is often difficult to develop fast algorithms via the dual problem. However, if the regularizer is strongly convex, the dual (or semi-dual) will be smooth, and standard gradient-based methods can be used. This is computationally tractable when the gradients of the dual can be computed efficiently [5]. Besides potentially expensive gradient computations, just as for Sinkhorn, lower regularization parameters will slow down the algorithm. Furthermore, regularizers that are not strongly convex cannot be handled in this framework. Therefore, we propose a different technique to solve (1) for a broad class of regularizers, including many non-smooth and non-strongly convex functions. Using the Douglas-Rachford splitting technique, we derive an algorithm with strong theoretical guarantees, that solves a range of practical problems rapidly and accurately. In particular, it efficiently handles regularizers that promote sparse transportation plans, in contrast to entropic regularization.

## 1.1 Contributions

We make the following contributions:

- We adapt the Douglas-Rachford splitting technique to regularized optimal transport, extending the recent DROT algorithm proposed in [27].

- Focusing on a broad class of regularizers which includes quadratic regularization and group lasso as special cases, we demonstrate global convergence guarantees and an accelerated local linear convergence rate. This extends the available theoretical results for related OT solvers and captures the behavior of the iterates observed in practice.

- We develop an efficient GPU implementation that produces high-quality optimal transport plans faster than the conventional Sinkhorn algorithm. We then show how the proposed solver can be used for domain adaption and to produce solutions of better quality up to 100 times faster than the state-of-the-art when implemented on GPU.

## 1.2 Related Work

The Sinkhorn algorithm is arguably one of the most popular OT solvers, and it can often find approximate solutions fast, even for large-scale OT problems. Many extensions and improvements have been proposed, including variations with improved numerical stability [36] and memory-efficient versions based on kernel operations [15]. However, to our best knowledge, no framework exists for general regularizers that enjoy comparable computational properties. As an example, a standard approach for group-lasso regularization is to linearize the regularization term and use Sinkhorn iteratively [9]. Although this approach is fairly generic, it requires that the transportation plan is recovered in every iteration, adding significant computational overhead. For strongly convex regularizers, such as quadratically regularized OT, several dual and semi-dual methods have been proposed, e.g. [5, 26], with stochastic extensions [37] and non-convex versions to deal with cardinality constraints [25]. However, these methods are significantly harder to parallelize, rendering them slower than Sinkhorn for larger problems [30]. Moreover, just as for Sinkhorn, both convergence rates and numerical stability properties deteriorate significantly with lower regularization parameters.

A promising research direction that has recently attracted interest in the community is to consider splitting methods as an alternative to Sinkhorn for OT problems. For instance, an accelerated primal-dual method for OT and Barycenter problems was proposed in [7], proximal splitting for a particular OT discretization was explored in [28], and an algorithm for unregularized OT based on Douglas-Rachford splitting was developed in [27]. The convergence of splitting methods is well-studied even for general convex problems [3]. For Douglas-Rachford splitting, tight global linear convergence

rates can be derived under additional smoothness and strong convexity assumptions [18]. Global linear rates have also been established for certain classes of non-smooth and non-strongly convex problems [41, 1]. Moreover, Douglas-Rachford splitting can benefit from the inherent sparsity of the problem at hand. For certain classes of problems, these algorithms can identify the correct sparsity pattern of the solution in finitely many iterations, after which a stronger local convergence rate starts to dominate [23, 31]. This paper contributes to this line of research, by introducing a splitting method for regularized OT with strong theoretical guarantees and unprecedented computational performance.

**Notation**

For any $X, Y \in \mathbb{R}^{m \times n}$, we let $\langle X, Y \rangle := \text{tr}(X^\top Y)$ and $\|X\|_F := \sqrt{\langle X, X \rangle}$. $\mathbb{R}_+^{m \times n}$ and $\mathbb{R}_-^{m \times n}$ denote the set of $m \times n$ matrices with non-negative entries and non-positive entries respectively. We let $\iota_S$ denote the indicator function over a closed convex set $S$, i.e $\iota_S(X) = 0$ if $X \in S$ and $\iota_S(X) = \infty$ if $X \notin S$, and the relative interior of a set $S$ is denoted $\text{relint}\, S$. The projection onto $\mathbb{R}_+^{m \times n}$ is denoted $[\cdot]_+$, which sets all negative entries of the inputted matrix to zero. The subdifferential of an extended-real valued function $h$ is denoted $\partial h(X)$, and its proximal operator is defined $\text{prox}_{\rho h}(x) = \text{argmin}_z h(z) + \frac{1}{2\rho} \|z - x\|^2$, where $\rho > 0$

## 2 Regularized Optimal Transport

We consider regularized OT problems on the form (1) where the function $h$ is *sparsity promoting* in the sense that its value does not increase if an element of $X$ is set to zero.

**Definition 2.1** (Sparsity promoting regularizers). *$h : \mathbb{R}^{m \times n} \to \mathbb{R} \cup \{+\infty\}$ is said to be sparsity promoting if, for any $X \in \mathbb{R}^{m \times n}$, $h(X) \geq h(X_s)$ for every $X_s \in \mathbb{R}^{m \times n}$ with $(X_s)_{ij} \in \{0, X_{ij}\}$.*

Notice that this function class does not necessarily induce sparsity. For instance $h = 0$, or $h = \|\cdot\|_F^2$ meet the conditions of Definition 2.1. Besides these two examples, the class of sparsity promoting functions include, but are not limited to, the following functions.

- $h(X) = \sum_{g \in \mathcal{G}} \|X_g\|_F$ (group lasso OT)
- $h(X) = \sum_{ij} X_{ij} \text{arcsinh}(X_{ij}/\beta) - (X_{ij}^2 + \beta^2)^{1/2}$ (hypentropic regularization)
- $h(X) = \sum_{ij} w_{ij} |X_{ij}|$, where $w_{ij} \geq 0$ (weighted $\ell_1$-regularization)
- $h(X) = \sum_{(ij) \in \mathcal{S}} \iota_{X_{ij}=0}(X)$ (constrained OT)

Conic combinations of sparsity promoting functions are also sparsity promoting. Moreover, many regularized OT problems, such as Gini-regularized OT [32] and regularizers on the form $\|X - A\|_F^2$, can be converted to only involve sparsity promoting regularizers. Note that the negative Shannon entropy $\sum X_{ij} \log X_{ij}$ is not sparsity promoting.

In this paper, we will develop a scalable algorithm for solving OT problems with sparsity-promoting regularizers. Besides strong theoretical guarantees, the algorithm is numerically stable and can be implemented efficiently on a GPU. Our approach builds on the recently proposed DROT algorithm for unregularized OT [27]. We extend the DROT algorithm to regularized OT problems, improve the theoretical convergence guarantees, and develop an extended GPU kernel. The resulting algorithm solves regularized OT problems faster, with higher precision, and in a more generic fashion than the state-of-the-art. To explain the algorithm, we review the splitting method for OT developed in [27].

### 2.1 Douglas-Rachford splitting for OT

Douglas-Rachford splitting is a technique for solving optimization problems on the form

$$\underset{x \in \mathbb{R}^n}{\text{minimize}} \ f(x) + g(x) \tag{2}$$

using the fixed point update $y_{k+1} = T(y_k)$, where

$$T(y) = y + \text{prox}_{\rho g}\left(2\text{prox}_{\rho f}(y) - y\right) - \text{prox}_{\rho f}(y) \tag{3}$$

and $\rho > 0$ is a stepsize parameter. If $y^\star$ is a fixed point of $T$ then $x^\star = \mathrm{prox}_{\rho f}(y^\star)$ is a solution to (2). Often a third iterate, $z_k$, is introduced, and the Douglas-Rachford iterations are expressed as

$$x_{k+1} = \mathrm{prox}_{\rho f}(y_k), \quad z_{k+1} = \mathrm{prox}_{\rho g}(2x_{k+1} - y_k) \quad y_{k+1} = y_k + z_{k+1} - x_{k+1}. \quad (4)$$

As long as $f$ and $g$ in (2) are closed and convex (possibly extended-real valued), and an optimal solution exists, the DR-splitting method converges to a solution (see, e.g. Theorem 25.6 in [3]). Under additional assumptions on $f$ and $g$, the method has even stronger convergence guarantees [18]. For a given problem that can be formulated on the form (2), there are typically many ways to partition the problem between $f$ and $g$. This must be done with care: a poor split can result in an algorithm with a higher per-iteration cost than necessary or one which requires more iterations to converge than otherwise needed. It is often difficult to achieve both simultaneously, which poses a challenging trade-off between the iterate complexity and the per-iteration cost of the resulting algorithm.

To facilitate the derivation of the algorithm, we let $\mathcal{X} = \{X \in \mathbb{R}^{m \times n} : X\mathbf{1}_n = p, \ X^\top \mathbf{1}_m = q\}$ and introduce the indicator functions $\iota_{\mathbb{R}_+^{m \times n}}$ and $\iota_{\mathcal{X}}$, that correspond to the constraints of (1). Recognizing that the projection onto $\mathcal{X}$ is tractable (by invoking recently derived formulas for matrix projections [4]) the authors of [27] proposed the splitting

$$f(X) = \langle C, X \rangle + \iota_{\mathbb{R}_+^{m \times n}}(X) \quad \text{and} \quad g(X) = \iota_{\mathcal{X}}(X)$$

and demonstrated that the update (3) can be simplified to

$$X_{k+1} = [Y_k - \rho C]_+, \quad Y_{k+1} = X_{k+1} + \phi_{k+1}\mathbf{1}_n^\top + \mathbf{1}_m \varphi_{k+1}^\top \quad (5)$$

where $\phi_k$ and $\varphi_k$ are vectors given via the iterative scheme:

$$r_{k+1} = X_{k+1}\mathbf{1}_n - p, \quad s_{k+1} = X_{k+1}^\top \mathbf{1}_m - q, \quad \eta_{k+1} = f^\top r_{k+1}/(m+n),$$
$$\theta_{k+1} = \theta_k - \eta_{k+1}, \quad a_{k+1} = a_k - r_{k+1}, \quad b_{k+1} = b_k - s_{k+1},$$

$$\phi_{k+1} = (a_k - 2r_{k+1} + (2\eta_{k+1} - \theta_k)\mathbf{1}_m)/n,$$
$$\varphi_{k+1} = (b_k - 2s_{k+1} + (2\eta_{k+1} - \theta_k)\mathbf{1}_n)/m. \quad (6)$$

The most expensive operations in (6) are the matrix-vector products in the $r$ and $s$ updates. Nonetheless, these operations can efficiently be parallelized on a GPU. Further, notice that $Y_k$ can be eliminated by substituting it in the $X$-update of (5), giving

$$X_{k+1} = [X_k + \phi_k \mathbf{1}_n^\top + \mathbf{1}_m \varphi_k^\top - \rho C]_+.$$

Hence, the splitting scheme only needs to update the transportation plan $X_k$ and the ancillary vectors and scalars in (6).

## 3  DR-splitting for regularized OT

When extending the DR-splitting scheme to the regularized setting, it is crucial to incorporate $h$ in one of the updates of (4) in a way that makes the resulting algorithm efficient. Specifically, if we let

$$f(X) = \langle C, X \rangle + \iota_{\mathbb{R}_+^{m \times n}}(X) + h(X), \text{ and } g(X) = \iota_{\mathcal{X}}(X), \quad (7)$$

the first update of (4) reads:

$$X_{k+1} = \mathrm{prox}_{\rho f}(Y_k) = \mathrm{prox}_{\rho h + \iota_{\mathbb{R}_+^{m \times n}}}(Y_k - \rho C).$$

This update is efficient given that $\mathrm{prox}_{\rho h + \iota_{\mathbb{R}_+^{m \times n}}}(\cdot)$ is easy to compute. This is indeed the case when $h$ is a sparsity promoting. We present the result in Lemma 3.1.

**Lemma 3.1.** *Let* $f(X) = \langle C, X \rangle + \iota_{\mathbb{R}_+^{m \times n}}(X) + h(X)$ *where* $h(X)$ *is sparsity promoting, closed, convex and proper when* $X \in \mathbb{R}_+^{m \times n}$. *Then* $\mathrm{prox}_{\rho f}(X) = \mathrm{prox}_{\rho h}([X - \rho C]_+)$.

Lemma 3.1 enables us to integrate the regularizer into the algorithm with only a minor update. Using the split of (7), the updated Douglas-Rachford algorithm, which we will refer to as RDROT, follows:

$$X_{k+1} = \mathrm{prox}_{\rho h}([Y_k - \rho C]_+), \quad Y_{k+1} = X_{k+1} + \phi_{k+1}\mathbf{1}_n^\top + \mathbf{1}_m \varphi_{k+1}^\top. \quad (8)$$

Under the assumption that $h$ is closed, convex, and proper over the feasible set of (1), Theorem 25.6 in [3] guarantees that RDROT converges to a solution of (1). Before we derive even stronger convergence properties of the algorithm, we describe a few specific instances in more detail.

**Quadratic Regularization.** Letting $h(X) = \frac{\alpha}{2}\|X\|_F^2$ yields

$$X_{k+1} = (1 + \rho\alpha)^{-1}[Y_k - \rho C]_+ \tag{9}$$

An attractive feature of quadratic regularization is that it shares similar limiting properties as entropic regularization when used in OT-divergences [14]. In contrast to entropically regularized OT, letting $\alpha \to 0$ does not lead to numerical instability in our method. Furthermore, the quadratic term makes the objective of (1) strongly convex which renders its optimal solution unique. This can be helpful when OT is used to define a loss function for e.g. training generative models [17] or adjusting for long-tailed label distributions [29], since it preserves some sparsity and renders the OT cost differentiable (see Section 3.3 for details).

**Group Lasso Regularization.** With $h(X) = \lambda \sum_{g \in \mathcal{G}} \|X_g\|_F$, where $\mathcal{G}$ is a collection of disjoint index sets, the RDROT update becomes

$$\bar{X}_{k+1} = [Y_k - \rho C]_+, \quad X_{k+1,g} = \left[1 - \frac{\lambda}{\|\bar{X}_{k+1,g}\|_F}\right]_+ \bar{X}_{k+1,g}, \quad g \in \mathcal{G}. \tag{10}$$

This regularizer has been used extensively for OT-based domain adaptation [10]. The rationale is that each unlabeled data point in the test (target) domain should only be coupled to data points of the training (source) domain that share the same label. This can be accomplished by organizing the data so that the rows of $X$ correspond to data points in the training domain and the columns to points in the test domain, and then using a sparsity-inducing group-lasso regularizer in (1). In this setting, $\mathcal{G}$ is a collection of sets, each specifying which columns correspond to a particular label. Consequentially, this regularizer will promote solutions that map each data point in the test domain to a single label.

## 3.1 Rate of convergence

It is well-known that DR-splitting finds an $\epsilon$-accurate solution in $O(1/\epsilon)$ iterations for general convex problems [21]. This is a significant improvement over Sinkhorn's iteration complexity of $O(1/\epsilon^2)$ [24], but the Sinkhorn iterations are very cheap to execute. A key contribution of [27] was the development of a GPU kernel that executes the DR updates so quickly that the overall method is faster than Sinkhorn for many problem instances. They also demonstrated a global linear convergence by exploiting the geometrical properties of LPs, but the convergence factors are very difficult to quantify or use for step-size tuning. As illustrated in Figure 1, and discussed in more detail below, neither the global linear rate nor the $1/k$-rate are globally tight. Instead, the stopping criterion decays as $O(1/k)$ during the initial steps of the RDROT algorithm, but then begins to converge at a fast linear rate. Hence, existing convergence analyses that neglect this local acceleration are unsatisfactory.

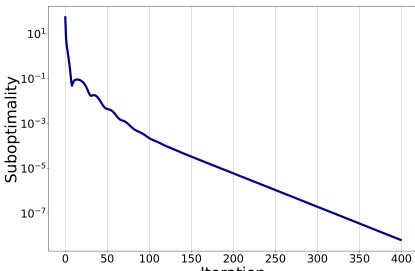 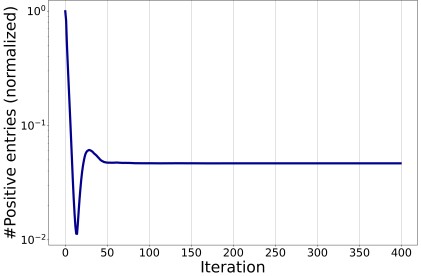

Figure 1: An illustration of the convergence behavior of the algorithm for a quadratically regularized problem. The stopping criterion and the number of positive entries were computed in every iteration. Notice that the iteration number at which the sparsity pattern stabilizes is aligned with when the linear convergence starts. In this example, the stepsize is chosen to make this effect more evident.

In this section, we will develop a convergence analysis for DR splitting on regularized OT problems that explains this observed behavior. Based on previous work on local convergence properties of

DR-splitting [23], we establish that our algorithm (8) identifies the true sparsity pattern of the solution in finitely many iterations. When the sparsity pattern is identified, a fast linear rate typically starts to dominate over the sublinear rate. To derive these guarantees, we need one additional assumption on the optimal solution of (1).

**Assumption 1.** *Let $Y^\star$ be a fixed point of* (3) *such that $Y_k \to Y^\star$, and let $X^\star = \text{prox}_{\rho f}(Y^\star) = \text{prox}_{\rho h}([Y^\star - \rho C]_+)$. We then assume that:*

$$\frac{1}{\rho}\left(Y^* - X^*\right) \in \text{relint}\,\partial f(X^\star). \tag{11}$$

Assumption 1 can be seen as a stricter version of the traditional optimality condition. It is a common assumption in analyses of active constraint identification that can be traced back to [6]. We consider this assumption to be fairly weak since it holds for most relevant OT problems, except for some very specific cost matrices. A more elaborate discussion is included in the supplementary material.

Under the regularity conditions of (11), we derive the following two convergence results.

**Theorem 1.** *Let $h(X)$ be sparsity promoting, convex, closed, and twice continuously differentiable for $X > 0$. If Assumption 1 holds, then there is a $K \geq 1$ such that for all $k \geq K$, $X_{k,\,ij} = 0$ if and only if $X_{ij}^\star = 0$.*

For our problem, this result implies that there is a $K$ after which the sparsity pattern does not change. To prove this, we first show that the sparsity-promoting and smoothness assumptions imply that $f$ and $g$ are partly smooth with respect to two manifolds and then invoke results presented in [23]. Moreover, Theorem 1 allows us to derive the following local convergence rate guarantees:

**Theorem 2.** *Assume that the conditions stated in Theorem 1 hold. If $h$ is locally polyhedral, then RDROT enjoys a local linear rate: When $k \geq K$, then*

$$\|X_{k+1} - X^\star\|_F \leq r^{k-K}\|Y_K - Y^\star\|_F, \quad r \in (0, 1).$$

*Moreover, the optimal rate is independent of the stepsize $\rho$.*

The regularizer $h$ is locally polyhedral, for instance, in the unregularized setting, and when weighted $\ell_1$ regularization is used (i.e. $h(X) = \sum_{ij} w_{ij}|X_{ij}|$, where $w_{ij} \geq 0$). Therefore, one can expect a sublinear rate until the correct sparsity pattern of the transportation plan is identified. From that point onward, a fast linear rate will dominate. The local rate holds for many other regularizers beyond locally polyhedral ones, which has also been observed in experiments (see, *e.g.*, [23] and [31]). However, when the local polyhedrality assumption is relaxed, one must account for the Hessian information of the regularizer, which makes it difficult to quantify a rate using this proof technique.

**Stepsize selection**   The stepsize implicitly determines how the primal residual descent is balanced against the descent in the duality gap. When the cost has been normalized so that $\|C\|_\infty = 1$, a stepsize that performs well regardless of $h$ is $\rho = 2(m + n)^{-1}$. This stepsize was proposed for DR-splitting on the unregularized problem in [27]. We use this stepsize in all our numerical experiments and observe that our method is consistently competitive without any additional tuning.

**Initialization**   A reasonable initialization of the algorithm is to let $X_0 = pq^\top$, and $\phi_0 = \mathbf{0}_m$, $\varphi_0 = \mathbf{0}_n$. However, when using the stepsize specified above, this will result in $X_k = \mathbf{0}_{m \times n}$ for $1 \leq k \leq N$, where $N = O(\max(m, n))$. By initializing $\phi_0 = (3(m + n))^{-1}(1 + m/(m + n))\mathbf{1}_m$, and $\varphi_0 = (3(m + n))^{-1}(1 + n/(m + n))\mathbf{1}_n$, one skips these first $N$ iterations, resulting in a considerable speed-up the algorithm, especially for larger problems. The derivation is added to the supplementary material. There is potential for further improvements (see e.g. [40]), since $\rho^{-1}\phi_k$, $\rho^{-1}\varphi_k$ are related to dual potentials. But for simplicity, we use this strategy throughout the paper.

**Stopping criteria**   One drawback of the Sinkhorn algorithm is that its theoretically justified suboptimality bounds are costly to compute from the generated iterates. This is usually addressed by only computing the primal residuals $\|X_k\mathbf{1}_m - p\|$, and $\|X_k^\top\mathbf{1}_n - q\|$ every $M$ iterations and terminate when both residuals are less than a chosen tolerance. In contrast, our algorithm computes the primal residuals in every iteration (see $r_k$ and $s_k$ in (6)), so evaluating the stopping criterion does not add any extra computations. Moreover, the iterates $\phi_k$ and $\varphi_k$ will, in the limit, be proportional to the optimal dual variables, and can hence be used to estimate the duality residual and duality gap in every iteration. For more details, we refer to the supplementary material.

## 3.2 GPU implementation

Similar to the approach developed in [27], the RDROT algorithm can be effectively parallelized on a GPU, as long as the regularizer is sparsity promoting and its prox-mapping is easy to evaluate. We have developed GPU kernels for RDROT with quadratic and group-lasso regularization, respectively[1]. Below, we provide an overview of how the DROT-kernel is adapted for the regularized case. More details of the implementation are included in the supplementary material.

For the GPU implementation, the main computation time is spent on updating the $X$ matrix as described in (8). Compared with DROT [27], the RDROT kernel is also required to evaluate the prox-mapping. Specifically, for the given regularizers, the following additional operations are needed:

- **Quadratic regularization:** Each element of $[Y_k - \rho C]_+$ needs to be rescaled with a constant to update $X_k$ (see (9)). This only requires a minor change and results in practically the same per-iteration-cost as for the unregularized case.

- **Group-lasso regularization:** The norm of each group, i.e $\bar{X}_{k+1,g}$, $g \in \mathcal{G}$ is required to the determine a group-specific scaling factor (see (10)). To this end, the following additional steps are required: (1) a reduction for gathering the square of the elements and computing the scale for the group; and (2) a broadcast to apply the scale to $\bar{X}_{k+1,g}$. For small problem sizes, the reduction and the broadcast can be done within a thread block via its shared memory. For large problem sizes, when the elements in one group can not fit in a single thread block, the results of the reduction are stored in global memory, and an additional kernel function is introduced to apply the scale for the group. This overhead increases the per-iteration cost a little, but we show in our experiments that it is still significantly faster compared to other methods.

The remaining parts of the algorithm are for updating vectors, which have much smaller time complexity than the $X$ matrix updates. These are handled with an ancillary kernel.

## 3.3 Backpropagation via Pytorch and Tensorflow wrappers

In many applications, e.g. [17, 35, 29], it is useful to use the solution of (1) to construct a loss function. Most notably, the optimal value of (1) is directly related to the Wasserstein distance between $p$ and $q$, which is a natural measure of the proximity between distributions with many advantageous properties (see [30] for an extensive treatment). When training generative models, for example, the objective is to transform a simple distribution into one that resembles the data distribution. It is therefore natural to use the Wasserstein distance to construct loss functions, see e.g [2]. But in order to backpropagate through such loss functions, one must differentiate the optimal value of (1) with respect to the cost matrix $C$ (which is parameterized by both a data batch and a generated batch of samples). Using Fenchel duality, we can show that the optimal solution $X^\star$ is a gradient (or Clarke subgradient) of the loss, regardless of the regularizer. By applying a similar argument to the dual problem, one can show that $\rho^{-1}(\phi^\star; \varphi^\star)$ is a reasonable gradient estimate of the OT cost with respect to the marginals $(p; q)$. Hence, one can use RDROT for both the forward and backward pass and do not need to lean on memory-expensive unrolling schemes or computationally expensive implicit differentiation techniques. We include further details on the derivation in the supplementary material. To facilitate using RDROT for DL, we wrapped our GPU OT solvers as PyTorch and TensorFlow extensions that feature fast automatic differentiation. The extensions can easily be integrated with networks that parameterize cost matrices or marginals. In a forward pass, our extension solves the regularized OT problem using the cost matrix and marginals given at the current iteration and stores the approximate solution $X_k^\star$ and $\rho^{-1}(\phi^\star; \varphi^\star)$ for the backward pass. The aforementioned technique is subsequently used in the backward pass to efficiently obtain an approximate gradient, which is used to rescale the gradients of the weights of the remaining network.

## 4 Numerical experiments

All experiments mentioned below are performed using a single NVIDIA Tesla V100 GPU.

---

[1]The implementation is publicly available at GitHub: https://github.com/WangZesen/Regularized-DROT

**GPU performance of RDROT with quadratic regularization**

To illustrate the numerical advantages of the algorithm, we compared the GPU implementation of RDROT for quadratically regularized OT (which we will refer to as QDROT) with a dual ascent method (L-BFGS), proposed in [5], which we implemented in PyTorch with all tensors loaded on the GPU. Although there are several other solvers for quadratic OT, such as the semi-dual method proposed in [5] and the semi-smooth Newton method from [26], we have omitted them from our experiments since their associated gradient/search direction computations scale poorly with data size and are difficult to parallelize. In our benchmark, we simulated 50 data sets of size $500 \times 500$ and $5000 \times 5000$, respectively. We generated cost matrices with $C_{ij} = \frac{1}{2}\|\mathbf{x}_i - \mathbf{x}_j\|_2^2$, where $\mathbf{x}_i, \mathbf{x}_j$ are simulated 2D Gaussian variables with random parameters specific for each data set. We used the quadratic regularization term $((m+n)\alpha/2)\|X\|_F^2$ with $\alpha$ ranging between $0.0005$ to $0.2$. For QDROT, we included $\alpha = 0$ for reference. We ran the algorithms for each dataset and regularization parameter until they reached an accuracy level of $0.0001$. For commpleteness, we also benchmark a PyTorch implementation of QDROT. The results are displayed in Figure 2, in which it is clear that our method is substantially faster than the L-BFGS method. Furthermore, when the GPU kernel is used, an even more significant speedup is achieved compared to the L-BFGS method. Further details and additional experiments with other dataset sizes are included in the supplementary material.

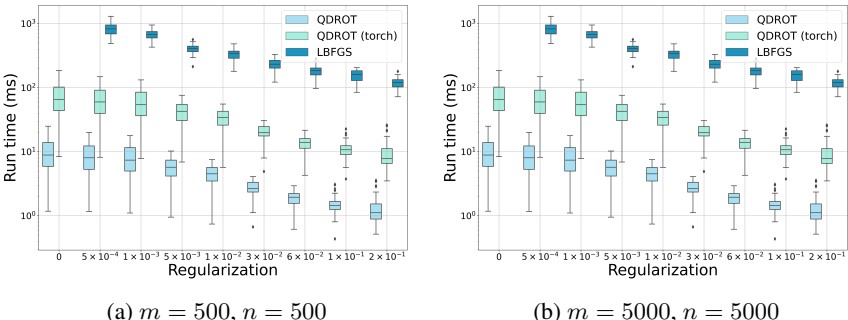

(a) $m = 500$, $n = 500$          (b) $m = 5000$, $n = 5000$

Figure 2: Comparison between the RDROT (QDROT) using the GPU kernel, a PyTorch version, and an L-BFGS method applied to the dual for different quadratic regularization parameters run on GPU. 50 datasets of two different sizes were simulated.

**Group Lasso and Domain Adaptation**

OT is the backbone of many domain adaption techniques since it provides a natural framework to estimate correspondences between a source domain and a target domain. These correspondences can subsequently be used to align the respective domains to improve the test accuracy. Specifically, for a given source data set $\{(\mathbf{x}_s^i, y^i)\}_{i=1}^m$ and a target set $\{\mathbf{x}_t^j\}_{j=1}^n$, where $\{y^i\}_{i=1}^m$ are data labels, we define the cost matrix elements $C_{ij} = d(\mathbf{x}_s^i, \mathbf{x}_t^j)$ using a positive definite function $d$. By computing a transportation plan $X$ associated with (1) and cost matrix $C$, one can adapt each data point in the source set into the target domain via $\mathbf{x}_{s,a}^i = \frac{1}{p_i}\sum_{j=1}^m X_{ij}\mathbf{x}_t^j$. It has been shown that group-lasso regularization can improve the adaptation [10] (cf. the discussion in §3). To handle the regularizer, it is customary to linearize it and iteratively solve updated OT problems with Sinkhorn [9, 10]. The current implementation of group-lasso OT in the Python OT package `POT` uses this approach [16]. Besides the computational cost of iteratively reconstructing transportation plans, entropic regularization tends to shrink the support of the adapted training data and hence underestimate the true spread in the target domain. This is thoroughly discussed in [15] and illustrated in Figure 3 where we adapt data sets with different regularizers. Of course, the shrinkage can partially be addressed by decreasing the regularization parameter, but this tends to slow down that algorithm and may even lead to numerical instability. To shed some light on this trade-off, we simulated $50$ datasets with two features, 1500 training samples, and 1000 test samples. The target domain was simulated via a random affine transformation of the source domain. Each set had 2 unique labels that were uniformly distributed among the instances. The labels in the test set were only used for validation. The cost $C_{ij} = \frac{1}{2}\|\mathbf{x}_s^i - \mathbf{x}_t^j\|^2$ was used and normalized so that $\|C\|_\infty = 1$. To compare the performance of

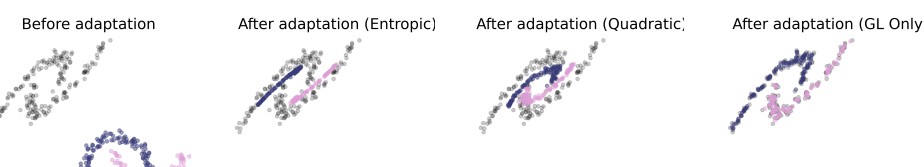

Before adaptation    After adaptation (Entropic)    After adaptation (Quadratic)    After adaptation (GL Only)

Figure 3: A toy example illustrating the effect strongly convex regularizations have on the support of the resulting adapted domains. Pink and purple points mark the labeled training set and the gray points the test domain. Notice how the Sinkhorn-based method, as well as the Quadratic Regularized method transfers the source domain samples towards the center of the target domain, while our approach transfers the samples in a way that matches the true variation of the test data

the algorithms, we computed the time taken to reach a tolerance of $10^{-4}$ and evaluted the Wasserstein distance between the adapted samples of each label and the corresponding samples of the test set. This means that the better the alignment, the lower the aggregated distance. For the method using entropic regularization, we varied the regularization parameters $0.001$ to $10$, and the group lasso regularization was set to $0.001$. The results are presented in Table 1. Indeed, increasing the entropic regularization parameter to speed up the conditional gradient methods will worsen the alignment between the source domain and the target domain, just as illustrated in Figure 3. We also note that our method consistently outperforms the alternative methods, both in terms of adaptation quality and in terms of time to reach convergence. Additional experiments with other problem sizes and regularization are consistent with Table 1, see the supplementary. In addition to the experiments

Table 1: Performance benchmark of RDROT for Group-Lasso regularization (GLDROT) against the conditional-gradient based Sinkhorn algorithm. Median and 10th and 90th percentile statistics are included. Our method is competitive in terms of both speed and adaption quality.

| | Reg. | | Runtime (s) $\downarrow$ | | | Agg. $W_2$ dist. $\downarrow$ | | |
|---|---|---|---|---|---|---|---|---|
| Method | Ent | GL | Median | $q10$ | $q90$ | Median | $q10$ | $q90$ |
| GLSK [10] | 1e-3 | 1e-3 | 3.77 | 3.68 | 8.90 | 0.311 | 0.0657 | 6.48 |
| GLSK [10] | 1e-1 | 1e-3 | 3.73 | 3.71 | 3.77 | 8.24 | 3.47 | 31.6 |
| GLSK [10] | 1e+2 | 1e-3 | 1.86 | 1.86 | 1.88 | 52.8 | 10.9 | 311 |
| GLDROT (ours) | - | 1e-3 | 0.0619 | 0.0475 | 0.0771 | **0.0576** | 0.0262 | 0.182 |
| GLDROT (ours) | - | 5e-3 | **0.0384** | 0.0306 | 0.0474 | 0.0879 | 0.0358 | 0.274 |

above, we compare the per-iteration costs for several OT solvers in the supplementary material. These strengthen our claim further that our algorithm achieves "lightspeed" even for large problem sizes.

**Training of generative models**

As an illustration of the efficiency of the GPU version of the algorithm, we use our PyTorch wrapper to implement the Minibatch Energy Distance [35], a loss function for training generative models via OT. The network consists of a generator, and a network that learns a cost function, and the OT-based Minibatch Energy distance to quantify the similarity between data batches and generated artificial samples. We performed experiments with image generation on the MNIST and CIFAR10 datasets, with some of the generated samples shown in Figure 4. The details of the experiments are included in the supplementary material. The resulting generator gives sharp images, and the training time was better or comparable with alternative frameworks, despite the low amount of regularization used.

## 5   Conclusions

The ability to solve large regularized optimal transport problems quickly has the potential to improve the performance of a wide range of applications, including domain adaption, learning of generative models, and more. In this paper, we have shown that the Douglas-Rachford splitting technique can handle a broad class of regularizers in an integrated and computationally efficient manner. Using a carefully chosen problem split, we derived an iterative algorithm that converges rapidly and reliably,

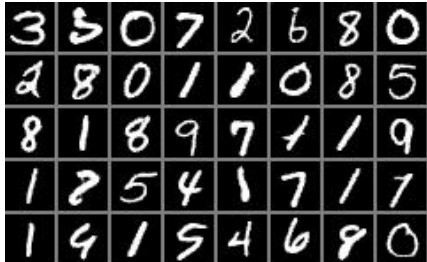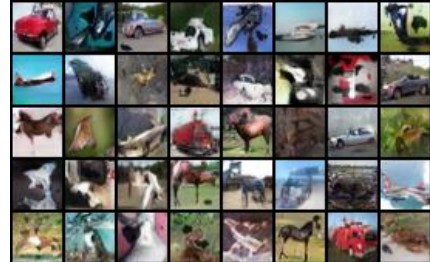

Figure 4: Generated samples by GANs trained on MNIST and CIFAR10 datasets with the Minibatch Energy Distance. The original sizes of samples are (28,28) and (32,32), respectively.

and whose operations are readily parallelizable on GPUs. The performance of our algorithm is supported by both theoretical arguments and extensive empirical evaluations. In particular, the experiments demonstrate that the resulting algorithms can be up to two orders of magnitude faster than the current state-of-the-art. We believe that this line of research has the potential to make regularized OT numerically tractable for a range of tasks beyond the ones discussed in this paper.

## Acknowledgement

This work was supported in part by the Knut and Alice Wallenberg Foundation, the Swedish Research Council, and the Wallenberg AI, Autonomous Systems and Software Program (WASP). The computations were enabled by resources provided by the Swedish National Infrastructure for Computing (SNIC), partially funded by the Swedish Research Council through grant agreement no. 2018-05973.

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

# A Theoretical results

## A.1 Proof of Lemma 3.1

To prove Lemma 3.1, the following result is helpful:

**Lemma A.1.** *Given that $h$ is sparsity promoting, closed, convex, and $\rho > 0$, then $X_{ij} = 0$ implies that $(\operatorname{prox}_{\rho h}(X))_{ij} = 0$. Moreover, if $X_{ij} > 0$, then $(\operatorname{prox}_{\rho h}(X))_{ij} \geq 0$.*

**Proof.** Let $Y = \operatorname{prox}_{\rho h}(X)$. By the definition of the proximal operator, every $Z \in \mathbb{R}^{m \times n}$ satisfies

$$h(Y) + \frac{1}{2\rho}\|Y - X\|_F^2 \leq h(Z) + \frac{1}{2\rho}\|Z - X\|_F^2$$

or

$$h(Y) + \frac{1}{2\rho}\left(\|Y - X\|_F^2 - \|Z - X\|_F^2\right) \leq h(Z).$$

Moreover, since $h$ is sparsity promoting $h(Z) \leq h(Y)$, and it holds that

$$\frac{1}{2\rho}\left(\|Y - X\|_F^2 - \|Z - X\|_F^2\right) \leq 0. \tag{12}$$

Let $Z$ be equal to $Y$ in all entries but the $(i, j)$th, which is set to zero. If $X_{ij} = 0$, then

$$\frac{1}{2\rho}\left(\|Y - X\|_F^2 - \|Z - X\|_F^2\right) = \frac{1}{2\rho}Y_{ij}^2$$

and invoking (12) gives that $Y_{ij} = 0$. If $X_{ij} > 0$, on the other hand, then

$$\frac{1}{2\rho}\left(\|Y - X\|_F^2 - \|Z - X\|_F^2\right) = \frac{1}{2\rho}\left((Y_{ij} - X_{ij})^2 - X_{ij}^2\right),$$

and (12) gives

$$(Y_{ij} - X_{ij})^2 \leq X_{ij}^2$$

which implies that $Y_{ij} \geq 0$. The proof is complete. $\qquad\square$

Lemma A.1 and Theorem 1 in [42] allows to prove the composition lemma quite easily. We end this section with the proof.

**Proof.** **(Lemma 3.1)** For $p(X) = \iota_{\mathbb{R}_+^{m \times n}}(X)$, it holds that

$$\operatorname{prox}_{\rho p}(X) = [X]_+$$

and

$$\partial p(X) = N_{\mathbb{R}_+^{m \times n}}(X) \supseteq \{0\}$$

for all $X \geq 0$, where $N_{\mathbb{R}_+^{m \times n}}(X) = \{Y \in \mathbb{R}^{m \times n} : Y_{ij} = 0 \text{ if } X_{ij} > 0, \ Y_{ij} \leq 0 \text{ if } X_{ij} = 0\}$.

By Lemma A.1, if $X_{ij} = 0$, then $(\operatorname{prox}_{\rho h}(X))_{ij} = 0$, and hence $(\partial p(X))_{ij} = (\partial p(\operatorname{prox}_{\rho h}(X)))_{ij}$. Further, for positive entries $X_{ij} > 0$, $(\partial p(X))_{ij} = \{0\}$, which implies that $(\partial p(\operatorname{prox}_{\rho h}(X)))_{ij} \supseteq (\partial p(X))_{ij}$. Altogether, this means that

$$\partial p(\operatorname{prox}_{\rho h}(X)) \supseteq \partial p(X).$$

Theorem 1 in [42] then gives

$$\operatorname{prox}_{\rho p + \rho h}(X) = \operatorname{prox}_{\rho h}\left(\operatorname{prox}_{\rho p}(X)\right) = \operatorname{prox}_{\rho h}\left([X]_+\right)$$

and

$$\begin{aligned}
\operatorname{prox}_{\rho f}(X) &= \operatorname{prox}_{\rho\langle C, \cdot\rangle + \rho p + \rho h}(X) \\
&= \operatorname{prox}_{\rho p + \rho h}(X - \rho C) \\
&= \operatorname{prox}_{\rho h}\left([X - \rho C]_+\right).
\end{aligned}$$

$\qquad\square$

## A.2 On the non-degeneracy condition of Asssumption 1

Assumption 1 is violated if $Y^*$ is a fixed point to equation (3) and $X^\star = \text{prox}_{\rho f}(Y^\star)$, but the regularity condition not fulfilled. That is, if

$$\frac{1}{\rho}\left(Y^* - X^*\right) \in \partial f(X^\star), \text{ while } \frac{1}{\rho}\left(Y^* - X^*\right) \notin \text{relint } \partial f(X^\star).$$

We can write these conditions more conveniently as

$$\frac{1}{\rho}\left(Y^* - X^*\right) \in \partial f(X^\star) \setminus \text{relint } \partial f(X^\star). \tag{13}$$

Let us now explore to what degree Assumption 1 can be seen as restrictive when it is invoked in Theorem 1. To this end, we also assume that $h$ is twice continuously differentiable for $X > 0$.

Recall that $f = \langle C, X \rangle + \iota_{\mathbb{R}^{m \times n}_+} + h(X)$ and note that the inclusion (13) cannot be satisfied if $X^* > 0$, since $\partial f(X^\star)$ is a singleton. We therefore focus on the case when $\partial f$ is set-valued, i.e. when $X^*_{ij} = 0$ for some $(i, j)$. Since $h(X)$ is differentiable for $X > 0$, we only need to consider the zero entries. If $X^*_{ij} = 0$, $\partial \iota_{\mathbb{R}^{m \times n}_+} = N_{\mathbb{R}^{m \times n}_+}$ implies that $(\partial \iota_{\mathbb{R}^{m \times n}_+} \setminus \text{relint } \partial \iota_{\mathbb{R}^{m \times n}_+})_{ij} = 0$ and thus that $\partial \iota_{\mathbb{R}^{m \times n}_+} \setminus \text{relint } \partial \iota_{\mathbb{R}^{m \times n}_+} = 0$. Consequentially, $\partial f(X^\star) \setminus \text{relint } \partial f(X^\star) = C + \partial h(X^*) \setminus \text{relint } \partial h(X^*)$, and

$$\frac{1}{\rho}\left(Y^* - X^*\right) = C + g, \text{ where } g \in \partial h(X^*) \setminus \text{relint } \partial h(X^*).$$

By the convergence of the algorithm, equation (5) in the main document implies that there exist two vectors $\phi^\star$ and $\varphi^\star$ such that $Y^\star - X^\star = \phi^\star \mathbf{1}_n^\top + \mathbf{1}_m \varphi^{\star \top}$. This gives that the cost is on the form

$$C = \rho^{-1}(\phi^\star \mathbf{1}_n^\top + \mathbf{1}_m \varphi^{\star \top}) - g.$$

In other words, it is only cost matrices with a specific (and rather particular) structure that violate Assumption 1. For instance, if $h$ is smooth, e.g. $h = \lambda \|\cdot\|_F^2$, Assumption 1 is only violated when the cost can be written as a sum of two given rank-1 matrices. This is rarely met in real applications.

## A.3 Proof of Theorem 1 and 2

To prove the theorem we need the notion of partial smoothness:

**Definition A.1** (Partial Smoothness). *A lower semi-continuous function $f : \mathbb{R}^n \to \mathbb{R}$ is said to be partly smooth at $x^\star \in \mathbb{R}^n$ with respect to the $C^2$-manifold $\mathcal{M} \subset \mathbb{R}^n$ if:*

(i) *The restriction of $f$ to $\mathcal{M}$: $f|_{\mathcal{M}}$ is $C^2$ around $x^\star$,*

(ii) *$\mathcal{T}_{\mathcal{M}}(x^\star) = (\text{par } \partial f(x^\star))^\perp$, where $(\text{par } \partial f(x^\star))^\perp$ denotes the orthogonal complement of the smallest subspace parallel to $\partial f(x^\star)$,*

(iii) *$\partial f(x)$ is continuous at $x^\star$ relative to $\mathcal{M}$.*

Let $X^\star$ be a solution to the OT problem that fulfills $X_k \to X^\star$, and let

$$\mathcal{M}_1 = \{X \in \mathbb{R}^{m \times n}_+ : X_{ij} = 0, \text{ if } X^\star_{ij} = 0, X_{ij} > 0 \text{ if } X^\star_{ij} > 0\}, \quad \mathcal{M}_2 = \mathcal{X}. \tag{14}$$

Recall that $f(X) = \langle C, X \rangle + \iota_{\mathbb{R}^{m \times n}_+}(X) + h(X)$, and $g(X) = \iota_{\mathcal{X}}(X)$. Since $h$ is assumed to be twice continuously differentiable over $\mathbb{R}^{m \times n}_{++}$, by the conditions in the theorem, $f$ is partly smooth with respect to $\mathcal{M}_1$. Further, since $g$ is constant over $\mathcal{M}_2$, it is also partly smooth over $\mathcal{M}_2$. The optimality conditions of the regularized OT problem are

$$0 \in \partial f(X^\star) - \frac{1}{\rho}\left(Y^* - X^*\right)$$

$$0 \in \partial g(X^\star) + \frac{1}{\rho}\left(Y^* - X^*\right). \tag{15}$$

Moreover, since $\partial g = N_{\mathcal{X}}$, and $\mathcal{X}$ is an affine subspace, $\partial g = \mathrm{relint}\, \partial g$. Therefore, Assumption 1 extends to the following the non-degeneracy condition:

$$
\begin{aligned}
0 &\in \mathrm{relint}\, \partial f(X^\star) - \frac{1}{\rho}\left(Y^* - X^*\right) \\
0 &\in \mathrm{relint}\, \partial g(X^\star) + \frac{1}{\rho}\left(Y^* - X^*\right).
\end{aligned}
\tag{16}
$$

With the partial smoothness and the non-degeneracy condition established, we have all the building blocks needed for proving Theorem 1 and 2.

**Proof.** (**Theorem 1 and 2**) By invoking the first order optimality condition associated with the first update in the DR-splitting scheme in (4), we get:

$$
X_{k+1} = \mathrm{prox}_{\rho f}\left(Y_k\right) \implies 0 \in \frac{1}{\rho}\left(X_{k+1} - Y_k\right) + \partial f(X_{k+1})
$$

or

$$
\frac{1}{\rho}\left(Y_k - X_{k+1}\right) \in \partial f(X_{k+1}).
\tag{17}
$$

Similarly, the second update in (4) is associated with the inclusion:

$$
Z_{k+1} = \mathrm{prox}_{\rho g}\left(2X_{k+1} - Y_k\right) \implies 0 \in \frac{1}{\rho}\left(Z_{k+1} - 2X_{k+1} + Y_k\right) + \partial g(Z_{k+1})
$$

or equivalently

$$
\frac{1}{\rho}\left(Y_k - X_{k+1}\right) - \frac{1}{\rho}\left(Y_{k+1} - Y_k\right) \in g(Z_{k+1}).
\tag{18}
$$

where we used the third equation of the DR-update, i.e $Y_{k+1} - Y_k = Z_{k+1} - X_{k+1}$. Notice how (17) and (18) relate closely to the optimality conditions (15). Now consider the function $F(X, Z) = f(X) + g(Z) - \rho^{-1}\langle X - Z, Y^* - X^*\rangle$. Since $X_{k+1} \in \mathbb{R}_+^{m \times n}$ and $Z_{k+1} \in \mathcal{X}$ for all $k \geq 1$, it holds that $F(X_{k+1}, Z_{k+1}) = \langle C, X_{k+1}\rangle + h(X_{k+1}) - \rho^{-1}\langle X_{k+1} - Z_{k+1}, Y^* - X^*\rangle$. Since $f + g$ are convex, closed, and attains a minimizer, Theorem 25.6 in [3] can be used to establish that $Y_k \to Y^\star$, when $k \to \infty$, where $Y^\star$ is a fixed point of the DR-update. In particular,

$$
\|Y_{k+1} - Y_k\| = \|(Y_{k+1} - Y^\star) - (Y_k - Y^\star)\| \leq \|Y_{k+1} - Y^\star\| + \|Y_k - Y^\star\| \to 0,
$$

which can be used to deduce that the third term of $F$ tends to zero as $k \to \infty$:

$$
|\langle X_{k+1} - Z_{k+1}, Y^* - X^*\rangle| \leq \|Z_{k+1} - X_{k+1}\|\|Y^* - X^*\| = \|Y_{k+1} - Y_k\|\|Y^* - X^*\| \to 0.
$$

Consequentially,

$$
F(X_{k+1}, Z_{k+1}) \to \langle C, X^\star\rangle + h(X^\star) = f(X^\star) + g(X^\star).
\tag{19}
$$

As $h$ is assumed to be convex, proper, and lower semicontinuous, we have that $h$ is subdifferentially continuous in its domain (including at $X^\star$) [33]. This gives

$$
\begin{aligned}
\mathrm{dist}\left\{0,\, \partial_X F(X_{k+1}, Z_{k+1})\right\} &= \mathrm{dist}\left\{0,\, \partial f(X_{k+1}) - \frac{1}{\rho}\left(Y^* - X^*\right)\right\} \\
&\leq \rho^{-1}\|(Y_k - Y^*) - (X_{k+1} - X^*)\| \\
&\leq \rho^{-1}\|Y_k - Y^*\| + \rho^{-1}\|X_{k+1} - X^*\| \\
&\leq 2\rho^{-1}\|Y_k - Y^*\| \\
&\to 0.
\end{aligned}
$$

Here, the first inequality follows from that $\mathrm{dist}\{x, S\} = \inf\{\|x - y\| : y \in S\} \leq \|x - y\|$, for all $y \in S$. The second is the Cauchy Schwarz inequality, and the third inequality follows from the non-expansiveness of proximal operators:

$$
\|X_{k+1} - X^*\| = \|\mathrm{prox}_{\rho f}\left(Y_k\right) - \mathrm{prox}_{\rho f}\left(Y^*\right)\| \leq \|Y_k - Y^*\|.
$$

Similarly,

$$
\begin{aligned}
\mathrm{dist}\left\{0,\, \partial_Z F(X_{k+1}, Z_{k+1})\right\} &= \mathrm{dist}\left\{0,\, \partial g(X_{k+1}) + \frac{1}{\rho}\left(Y^* - X^*\right)\right\} \\
&\le \rho^{-1}\|(X_{k+1} - X^*) - (Y_k - Y^*) - (Y_{k+1} - Y_k)\| \\
&\le 2\rho^{-1}\|Y_k - Y^*\| + \rho^{-1}\|Y_{k+1} - Y_k\| \\
&\to 0.
\end{aligned}
$$

Hence, $\mathrm{dist}\left\{0,\, \partial F(X_{k+1}, Z_{k+1})\right\} \to 0$. This together with (16) and (19), and the fact that $f$ is partially smooth at $X^\star$ with respect to $\mathcal{M}_1$, we apply Theorem 5.3 in [20] which proves the theorem. To prove Theorem 2, one can directly invoke Theorem 5.6 in [23]. $\qquad\square$

## B  Initialization

When using the initialization $X_0 = pq^\top$, $\phi_0 = \mathbf{0}_m$, and $\varphi_0 = \mathbf{0}_n$, and the default stepsize, the updates in (6) gives that

$$
\begin{aligned}
X_k &= \mathbf{0}_{m \times n} \\
\phi_k &= (k+1)n^{-1}(p + (m+n)^{-1}) \\
\varphi_k &= (k+1)m^{-1}(q + (m+n)^{-1})
\end{aligned}
$$

when $k = 1, 2, \ldots N$, where

$$
N = \min_{ij}\lceil C_{ij} mn(m+n)^{-1}(mp_i + nq_j + 1)^{-1} - 1\rceil = O(\max(m,n)).
$$

When the cost is normalized, we can use the rough approximation, $C_{ij} \sim 1$, $p_i \sim m^{-1}$, and $q_j \sim n^{-1}$, to get the simple initialization strategy: $X_0 = \mathbf{0}_{m \times n}$, $\phi_0 = (3(m+n))^{-1}(1 + m/(m+n))\mathbf{1}_m$, and $\varphi_0 = (3(m+n))^{-1}(1 + n/(m+n))\mathbf{1}_n$, that skips these first $N$ iterations.

## C  Derivation of Dual and Stopping criterion

A simple calculation reveals that $g(X) = \iota_\mathcal{X}(X)$ has the Fenchel conjugate

$$
g^*(U) = \begin{cases} \langle U, pq^\top\rangle, & \text{if } U = \mu\mathbf{1}_n^\top + \mathbf{1}_m\nu^\top \\ \infty, & \text{otherwise.} \end{cases}
$$

Hence, any feasible dual variable is on the form $U = \mu\mathbf{1}_n^\top + \mathbf{1}_m\nu^\top$, for which $g^*(U) = p^\top\mu + q^\top\nu$. Further, the conjugate of $f(X) = \langle C, X\rangle + \iota_{\mathbb{R}_+^{m \times n}}(X) + h(X)$ can be expressed as

$$
\begin{aligned}
f^*(U) &= \sup_X \langle U, X\rangle - f(X) \\
&= \sup_{X \ge 0} \langle U - C, X\rangle - h(X) \\
&\le \sup_{X \ge 0} \langle [U - C]_+, X\rangle - h(X) \\
&\le \sup_X \langle [U - C]_+, X\rangle - h(X) \\
&= h^*([U - C]_+).
\end{aligned}
$$

Moreover, since $h$ is sparsity promoting,

$$
\begin{aligned}
h^*([U - C]_+) &= \sup_X \langle [U - C]_+, X\rangle - h(X) \\
&\le \sup_X \langle [U - C]_+, [X]_+\rangle - h([X]_+) \\
&= \sup_{X \ge 0} \langle [U - C]_+, X\rangle - h(X) \\
&= f^*(U),
\end{aligned}
$$

meaning that $f^*(U) = h^*([U - C]_+)$. This gives the dual problem

$$\underset{U = \mu \mathbf{1}_n^\top + \mathbf{1}_m \nu^\top}{\text{maximize}} \; -f^*(-U) - g^*(U) = -p^\top \mu - q^\top \nu - h^*([-\mu \mathbf{1}_n^\top - \mathbf{1}_m \nu^\top - C]_+). \qquad (20)$$

To relate the iterates to the optimality condition, we define $X_{k+1} = \text{prox}_{\rho f}(Y_k)$, $Z_{k+1} = \text{prox}_{\rho g}(2X_{k+1} - Y_k)$, and $U_k := (Y_{k-1} - X_k)/\rho$ and observe that

$$\rho^{-1}(Z_{k+1} - X_{k+1}) = \rho^{-1}(X_{k+1} - Y_k - \phi_k \mathbf{1}_n - \mathbf{1}_m \varphi_k^\top) = -U_{k+1} - (\rho^{-1}\phi_k)\mathbf{1}_n - \mathbf{1}_m(\rho^{-1}\varphi_k)^\top.$$

By theorem 25.6 in [3], $Z_{k+1} - X_{k+1} \to 0$, and $U_{k+1} \to U^\star$ where $U^\star$ is a solution to (20). Hence $(\rho^{-1}\phi_k)\mathbf{1}_n + \mathbf{1}_m(\rho^{-1}\varphi_k)^\top \to U^*$, meaning that $\rho^{-1}\phi_k$ and $\rho^{-1}\varphi_k$ will correspond to $\mu$ and $\nu$ in (20) respectively. Using that $r_k$ and $s_k$ in (6) are directly related to the primal residual, we can derive the following stopping criteria:

$$r_{\text{primal}} := \max(\|X_k \mathbf{1}_n - p\|, \|X_k^\top \mathbf{1}_m - q\|) = \max(\|r_k\|, \|s_k\|),$$
$$\text{gap} := \langle C, X_k \rangle + h(X_k) - \rho^{-1}(p^\top \phi_k + q^\top \varphi_k) + h^*(\rho^{-1}([\phi_k \mathbf{1}_n^\top + \mathbf{1}_m \varphi_k^\top - \rho C]_+)$$
$$= \langle C, X_k \rangle - \rho^{-1}(p^\top \phi_k + q^\top \varphi_k) + \Delta_h(X_k, \phi_k, \varphi_k),$$

where $\Delta_h$ denotes the duality gap deviation compared to the unregularized problem. Further, notice that by letting $\bar{X}_{k+1} = [X_k + \phi_k \mathbf{1}_n^\top + \mathbf{1}_m \varphi_k^\top - \rho C]_+$, we can express $[\phi_k \mathbf{1}_n^\top + \mathbf{1}_m \varphi_k^\top - \rho C]_+ = [\bar{X}_{k+1} - X_k]_+$, which facilitates evaluating the duality gap. Notice that $h^*$ may not be finite. In such settings, the dual is maximized over its effective domain, resulting in the addition of dual constraints. To account for these, one must include a dual residual that measures the constraint violation. We include some examples of stopping criteria in the following table.

Table 2: Termination criteria for a selection of regularizers. Here we let $\| \cdot \|$ denote an arbitrary norm that is sparsity promoting.

| $h$ | $\Delta$ | $r_{\text{dual}}$ |
|---|---|---|
| $0$ | $0$ | $\|[X_{k+1} - X_k]_+\|_F$ |
| $\|X\|_F^2$ | $\|[(1+\rho)X_{k+1} - X_k]_+\|_F^2$ | $0$ |
| $\|X\|$ | $0$ | $\|\text{prox}_h\left([\bar{X}_{k+1} - X_k]_+\right)\|_F$ |

## D  Backpropagation through regularized OT-costs

We let $h_c = h + \iota_{\mathbb{R}_+^{m \times n}} + \iota_{\mathcal{X}}$, and define the OT-cost as follows: $\text{OT}_h(C) = \inf_X \langle C, X \rangle + h_c(X)$. Note that OT cost is equal to the optimal value of the OT problem (1), i.e. if $X^\star$ is a solution to (1), then $\text{OT}_h(C) = \langle C, X^\star \rangle + h_c(X^\star)$. The OT cost is also closely related to the Fenchel conjugate of $h_c$, as $\text{OT}_h(C) = -\bar{h}^*(-C)$, and hence $\bar{h}^*(-C) + \bar{h}(X^\star) = \langle -C, X^\star \rangle$. Invoking subdifferential properties of the Fenchel conjugate gives that $X^\star \in \partial \bar{h}^*(-C)$. Further

$$\partial_C(-\text{OT}_h(C)) = \partial_C h_c^*(-C) \ni -X^\star.$$

In particular, whenever the solution is unique, such as when $h$ is strongly convex, then $\nabla_C \text{OT}_h(C) = X^\star$. Therefore, using the transportation plan estimated via RDROT is a reasonable gradient estimate of the OT cost.

## E  GPU Implementation

### E.1  GPU Architecture Basics

Before introducing the detailed design of the GPU kernels, we recall some basic properties of modern GPU architectures. Since the kernel design is based on CUDA, which is compatible with NVIDIA GPUs, we focus on the architecture of NVIDIA GPUs. Nevertheless, many programming concepts, hardware architecture decisions, and performance bottlenecks are shared with other GPU designs.

### E.1.1 Thread Organization

The fundamental unit of computation in NVIDIA GPUs is the Streaming Multiprocessor (SM). It is responsible for executing the parallel computations and managing the resources on the GPU. Each SM consists of multiple CUDA cores, a shared memory, registers, and other components that enable efficient parallel processing. However, the concepts introduced below also apply to other modern GPU architecture that supports the SIMT execution model (like the AMD and Intel GPUs).

The parallel computations are organized in threads, grouped according to the following hierarchy:

1. The *thread* is the basic unit of code. All threads execute the same code, but they are endowed with an ID that can be used to parameterize memory access and control decisions.

2. A *block* is a group of threads that execute together on an SM in the GPU. Threads within a block can cooperate and communicate using shared memory.

3. The *grid* refers to the entire set of blocks that will be executed on the GPU. It represents the overall organization of parallel computation.

The smallest unit of thread execution on an NVIDIA GPU is the *warp*. It consists of 32 threads that are scheduled and executed together on a single SM. All threads within a warp execute the same instruction at the same time (but on different data, parameterized by the thread and block IDs). The reduction within the warp is highly optimized and efficient.

### E.1.2 Memory Hierarchy

NVIDIA GPUs, such as the NVIDIA Tesla V100, have more than eight types of memory including global memory, shared memory, registers, texture memory, constant memory, etc. We only focus on the first three memory types, since they are the most relevant to the kernel design in the next section.

1. On NVIDIA GPUs, the *global memory* refers to the main memory available on the GPU. It is a high-speed memory space used for storing data and instructions that are accessible by all the CUDA cores in the GPU. The global memory is the entry for exchanging data with the system RAM (random access memory). The global memory lies in the highest level of the memory hierarchy and has the largest capacity and the slowest access speed. NVIDIA Tesla V100 has 32GB of global memory. An important feature of global memory is coalesced memory access, which will be discussed below and used in Section E.2.1.

2. The *shared memory* is a low-latency memory space, physically located on the GPU chip. The shared memory is used for data sharing within a block, reducing memory access times significantly compared with using the global memory. The access speed of the shared memory is roughly 100x faster than the uncached global memory. Other characteristics are that it is allocated for and shared between threads in the same block, and that its capacity is limited compared with the global memory. For example, NVIDIA Tesla V100 with compute capability 7.0, has 96KB of shared memory per SM and 80 SMs in total.

3. The *register* memory of NVIDIA GPUs is the fastest memory with the lowest latency. Registers are private to each thread and are not shared among other threads or blocks. They are used for temporary data storage and computation in a single thread. The access speed for registers is faster than the shared memory, but not by orders of magnitude. Register memory also has limited capacity. NVIDIA Tesla V100 has a limit of 64K 32-bit registers per SM.

### E.1.3 Coalesced Memory Access

An important feature of global memory management is coalesced memory access. It is a memory access pattern that maximizes memory bandwidth and improves memory performance in parallel computing, particularly on GPUs. It refers to the way threads in a warp access memory in a contiguous and aligned manner, minimizing memory transactions and maximizing data transfer efficiency.

Depending on the architecture of the GPUs, the device can load or write the global memory via a 32-byte, 64-byte, or 128-byte transaction that is aligned with their sizes, respectively. If the threads in a warp access a continuous and aligned memory block with the size of 128-byte (this could be 32 4-byte single-precision floats, for example), it only costs one transaction. If the continuous memory block is misaligned with the 128-byte pattern, it costs two transactions. If the memory accessed

is strided, which means no two elements are in the same 128-byte memory block, it will cost 32 transactions, even though it accesses the same amount of memory as in the coalesced case.

## E.2 GPU Kernel Design

Our GPU kernels are the enhanced and optimized versions of the code for unregularized OT that accompanies the paper [27], available on Github[2]. In the following subsections, we describe our improvements of the basic kernel, and detail the extensions that we have made to implement quadratic and group-lasso regularization. We will also present run-time measurements to demonstrate that our kernels execute even faster than the "light-speed" per-iteration times of Sinkhorn.

### E.2.1 General Optimization

Similar to the kernel described in [27], the main kernel is designed in the way that each thread block contains 64 threads and is responsible for updating a sub-matrix with size $(64, 64)$ in $X$. However, for small problem sizes where the required number of blocks may be smaller than the number of SMs, this design may cause low utilization of the GPU since not all of the SMs are used. To improve the utilization for small problem sizes, we adapt the number of columns assigned to each block to the problem size, reducing the per-block work load and increasing the SM utilization.

Careful management of the global memory access is critical to the performance of CUDA applications. In the kernel from [27], uncoalesced memory access happens when the number of rows is not a multiple of 32. For example, when the number of rows is 127 and the block tries to update the second column of $X$, the threads in the block will access 64 floats. However, the continuous block is not aligned with the 128-byte block structure, because the first element of the second column is in the same block as the last 31 elements in the first column. In this case, accessing 64 single-precision floats will require three 128-byte transactions while it could be two 128-byte transactions in an optimal solution. This can lead to strange situations where decreasing the problem size by one increases the per-iteration runtime significantly (6-20%). To alleviate the problem and to make full use of the coalesced memory access, we updated the kernel from [27] to handle shifted columns that match the 128-byte memory blocks. An illustration of the update is in Figure 5 where the old working area of a block from [27] is marked with "Planned Block Work Area" and the updated working area of a block is marked with "Actual Block Work Area". The comparison of the per-iteration runtime before and after the optimization in Table 3 shows that the update solves the problem of uncoalesced memory access, and that the implementation performs more predictably on various problem sizes.

Table 3: Comparison of the per-iteration runtime before and after the update for coalesced memory access. Note that the focus here is on the relative change in the per-iteration runtime after decreasing the problem size by one. The improvement in the per-iteration runtime under the same problem size is caused by other optimization techniques.

| Problem Size | Runtime Before Update | Runtime After Update |
|---|---|---|
| $4096 \times 4096$ | 0.3577 | 0.3453 |
| $4095 \times 4095$ | 0.3812 (+6.58%) | 0.3417 (-1.04%) |
| $8192 \times 8192$ | 1.281 | 1.152 |
| $8191 \times 8191$ | 1.548 (+20.80%) | 1.153 (+0.10%) |
| $10240 \times 10240$ | 1.916 | 1.808 |
| $10239 \times 10239$ | 2.189 (+14.24%) | 1.781 (-1.52%) |

### E.2.2 Quadratically Regularized DROT

Similar to the implementation in [27], the matrices $X$ and $C$ are stored in the global memory in column-major order. For the problem sizes that we target, the global memory is the only one that can fit the matrices. In addition, the performance penalty of storing them in a slower memory is limited since each element only needs to be loaded once for every update. As the update of $X$ is done column by column, the auxillary variables $\phi$ and $\varphi$ can be shared and reused among the threads in a block. To speedup the memory access, we load the required elements of $\phi$ and $\varphi$ into shared memory at the beginning of the main kernel.

---

[2]The implementation is open-source and available at https://github.com/vienmai/drot

Figure 5 illustrates how the update of one sub-column is done in a thread block of the quadratically regularized DROT kernel. More specifically, an update is comprised of the following steps

(1) Load the corresponding elements of $\phi$ and $\varphi$ from the shared memory.
(2) Load the corresponding elements of $X$ and $C$ from the global memory. Compute $[Y_{ij} - \rho C_{ij}]_+$ and $C_{ij} \cdot X_{ij}$. For the quadratically regularized case, we can directly update $X_{ij}$ with $Y_{ij}$ after multiplying it with the scale $1/(1 + \rho\alpha)$.
(3) Perform in-warp reduction to compute the column sum and the objective value in the block.
(4) Use atomic operations to accumulate the results to the corresponding row and column sums.
(5) This step is only used by the group-lasso kernel, introduced in Section E.2.3.

Similar to the implementation by [27], the iterations are divided into even iterations and odd iterations.

1. For the even iterations, the kernel updates $X_{k+1}$ as
$$X_{k+1} = \text{prox}_{\rho h}\left([X_k + \phi_{k+1}\mathbf{1}_n^\top + \mathbf{1}_m\varphi_{k+1}^\top - \rho C]_+\right) - \rho C$$

2. For the odd iterations, it updates $X_{k+1}$ as
$$X_{k+1} = \text{prox}_{\rho h}\left([X_{k+1} + \phi_{k+1}\mathbf{1}_n^\top + \mathbf{1}_m\varphi_{k+1}^\top]_+\right)$$

In this way, the kernel only needs to access the matrix $C$ once every second iteration.

Except for the general optimizations introduced in Section E.2.1, the only difference from the basic kernel is step (2) which additionally multiplies the result with a scalar. This minor change introduces negligible changes in the per-iteration runtime.

### E.2.3 Group-lasso Regularized DROT

The kernel for group-lasso regularized DROT is slightly more involved, but also follows the five-step procedure illustrated in Figure 5. More specifically, it performs the following steps:

(1) Load the corresponding elements of $\phi$ and $\varphi$ from the shared memory.
(2) Load the corresponding elements of $X$ and $C$ from the global memory. Compute $[Y_{ij} - \rho C_{ij}]_+$ and $C_{ij} \cdot X_{ij}$. Compared with the quadratically regularized DROT, the kernel cannot update $X_{ij}$ until the scale of the group is ready in step (5).
(3) Perform in-warp reduction. For the group-lasso regularized case, we also reduce the sum of $[Y_{ij} - \rho C_{ij}]_+^2$ to compute the norm of the group.
(4) Accumulate the results to the corresponding sum of the row, the sum of the column, and the sum of the squared elements within the group using atomic operations.
(5) Compute the scale for the group and apply it to the computed $[Y_{ij} - \rho C_{ij}]_+$ so as to update $X_{ij}$. (For large problem sizes, where a thread block can only access part of the group, the process of reduction and broadcast is done in an additional kernel function).

Compared with the quadratically regularized case, the group-lasso regularized DROT requires an additional round for reduction and broadcast within the group. Moreover, for large problem sizes, when the intermediate results for the group do not fit in the shared memory, it is inevitable to access the matrix $C$ in every iteration. Both factors lead to an increase in the per-iteration runtime of the Group-lasso regularized DROT, which ends up being roughly twice as long as that of the quadratically regularized DROT. However, as we will show in the next section, its per-iteration runtime is still comparable with the Sinkhorn-Knopp algorithm.

### E.2.4 Per-iteration Runtime

The comparison of the per-iteration runtime among all methods is given in Figure 6. For QDROT and the Sinkhorn-Knopp algorithm, 12 random samples are generated with problem sizes from 100 to 10000. For GLDROT, 12 random samples are generated with problem sizes from 100 to 10000 and with 2 and 4 classes, respectively. Note that the Sinkhorn-based implementation of group-lasso OT in POT [16] does not return the number of iterations, nor reports the per-iteration run-time. This makes

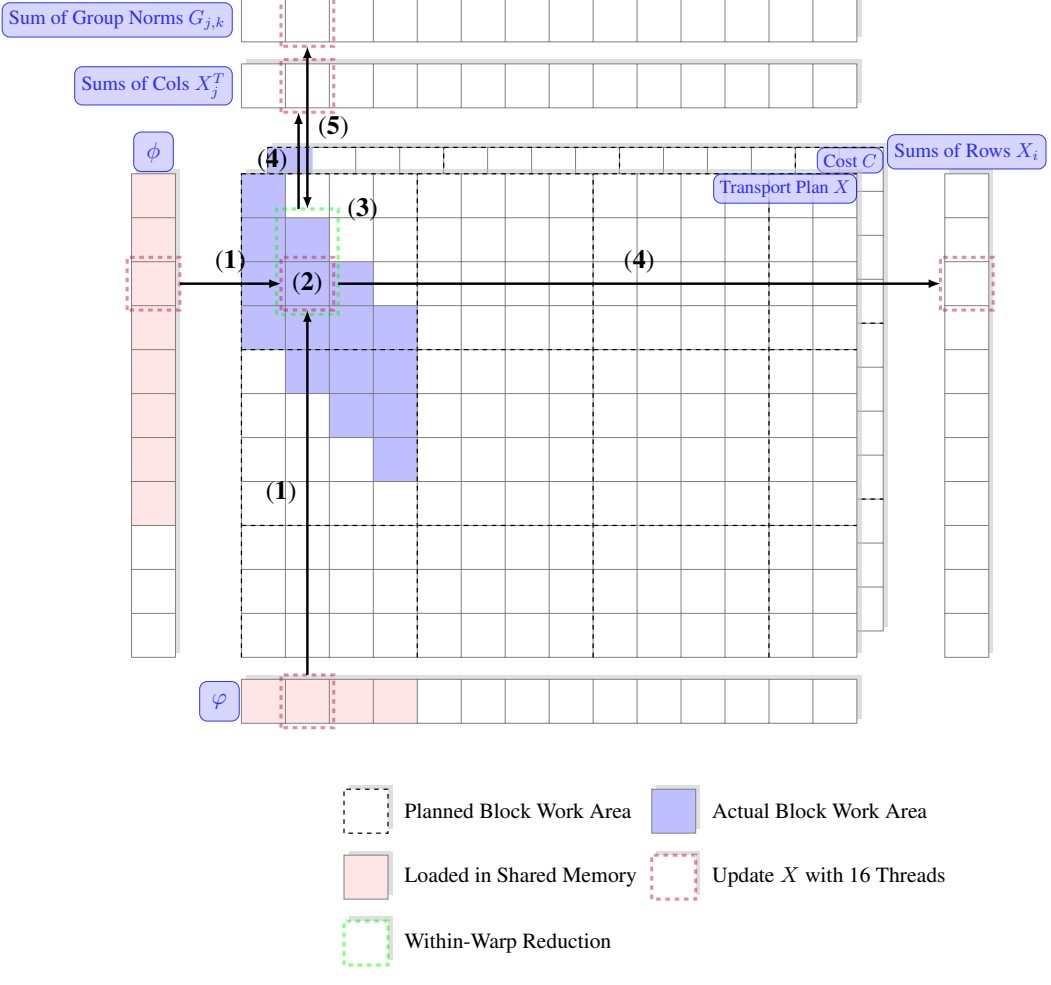

Figure 5: GPU Kernel Design

it impossible to estimate the corresponding run-time numbers for the Sinkhorn-based group-lasso OT [11]. We have therefore excluded this algorithm here, but expect it to have larger (and possibly significantly larger) per-iteration runtime than the Sinkhorn-Knopp algorithm [11].

To make a fair comparison between the Sinkhorn-Knopp algorithm and ours, we cover four variants of the Sinkhorn-Knopp algorithm in the comparison. The implementation of Sinkhorn-Knopp in POT [16] computes the primal residual once every 10 iterations, since this operation is relatively costly compared with the iteration itself. Moreover, to increase the numerical stability of the algorithm and to improve the quality of the results with smaller weights of the entropic regularizer, the Sinkhorn-Knopp algorithm is improved by moving the computation to log scale. The decision to compute the primal residual or not, and to perform the computations in linear or logarithmic scale leads to the four distinct versions of the Sinkhorn-Knopp algorithm that we evaluate. Since our methods work directly on $X$ we let them compute the objective value and the primal residual in every iteration.

The results show that our QDROT implementation has roughly the same per-iteration runtime as the standard Sinkhorn-Knopp algorithm. The per-iteration runtime of the Sinkhorn-Knopp algorithm in log scale is roughly 4.3-9.4x longer than QDROT and is roughly 2.5-4.5x longer than GLDROT.

# F   Additional Experiments

This section describes additional experiments with RDROT using quadratic and group-lasso regularization on more datasets of various sizes and using different hyperparameters. The results are

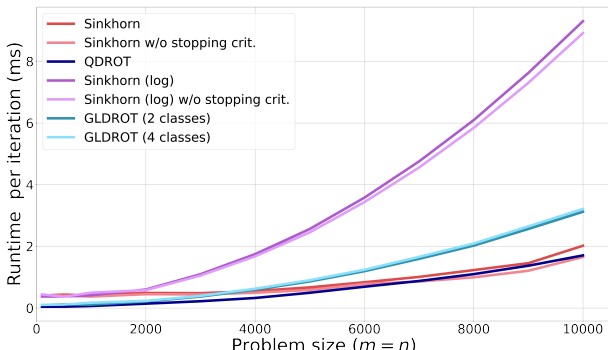

Figure 6: Comparison of the per-iteration cost for OT solvers. We compare the Sinkhorn-Knopp Algorithm [11], and a log domain variant (with and without stopping criterion computations every 10th iteration), against QDROT (ours): Quadratically regularized DROT, and GLDROT (ours): Group-lasso regularized DROT.

consistent with the experiments that could be included in the main document and demonstrate significant advantages of our method compared to the state-of-the-art.

**Quadratic regularization** In the main manuscript, we compared our solver with an L-BFGS method applied to the dual problem. Two problem sizes were considered: $m = 1000$, $n = 1000$, and $m = 2000$, $n = 3000$. In Figure 2, we rerun the benchmark on 4 additional problem sizes. Notice that all results are consistent with those presented in the paper, showing that our results generalize to more settings beyond the ones presented in the paper.

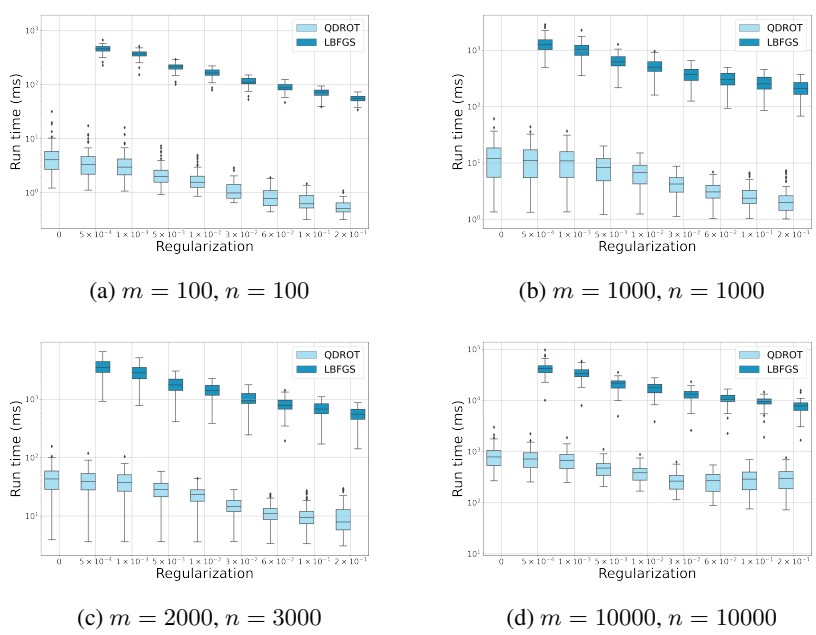

(a) $m = 100$, $n = 100$

(b) $m = 1000$, $n = 1000$

(c) $m = 2000$, $n = 3000$

(d) $m = 10000$, $n = 10000$

Figure 7: Additional experiments of the RDROT (QDROT) and the dual L-BFGS method comparison for different quadratic regularization parameters and problem sizes. 50 datasets of an additional 4 problem sizes were simulated.

**Group Lasso regularization** To establish the computational advantages of RDROT applied to the Group-Lasso problem, we benchmarked it against Sinkhorn-based solver on simulated datasets of size $m = 1000$, $n = 1500$, for a selection of hyperparameters. It Table 4,5,6, we rerun the benchmark on more problem sizes and hyperparameters, to test if the results generalize. Indeed, GLDROT consistently outperforms the state-of-the-art in all considered problems.

Table 4: GL experiments with $n = 500$ and $m = 500$

| Method | Reg. Ent | Reg. GL | Runtime (s) ↓ Median | Runtime (s) ↓ $q10$ | Runtime (s) ↓ $q90$ | Agg. $W_2$ dist. ↓ Median | Agg. $W_2$ dist. ↓ $q10$ | Agg. $W_2$ dist. ↓ $q90$ |
|---|---|---|---|---|---|---|---|---|
| GLSK | 1e-3 | 1e-6 | 1.26 | 1.24 | 2.98 | 0.35 | 0.084 | 4.96 |
|  | 1e-3 | 5e-4 | 1.23 | 1.22 | 2.93 | 0.351 | 0.0841 | 4.96 |
|  | 1e-3 | 5e-2 | 1.35 | 1.23 | 3.57 | 0.377 | 0.092 | 4.96 |
|  | 1e-2 | 1e-6 | 1.21 | 1.09 | 2.63 | 1.22 | 0.668 | 5.48 |
|  | 1e-2 | 5e-4 | 1.18 | 1.07 | 2.61 | 1.22 | 0.668 | 5.48 |
|  | 1e-2 | 5e-2 | 1.2 | 1.08 | 2.63 | 1.22 | 0.668 | 5.57 |
|  | 1e-1 | 1e-6 | 1.04 | 1.04 | 1.06 | 7.83 | 3.27 | 30.6 |
|  | 1e-1 | 5e-4 | 1.05 | 1.05 | 1.07 | 7.83 | 3.27 | 30.6 |
|  | 1e-1 | 5e-2 | 1.05 | 1.04 | 1.06 | 7.83 | 3.27 | 30.6 |
|  | 1. | 1e-6 | 1.03 | 0.517 | 1.04 | 45.3 | 9.74 | 212 |
|  | 1. | 5e-4 | 1.02 | 0.512 | 1.03 | 45.3 | 9.74 | 212 |
|  | 1. | 5e-2 | 1.03 | 0.514 | 1.03 | 45.3 | 9.74 | 212 |
| GLDROT |  | 1e-6 | 0.0924 | 0.0519 | 0.16 | 0.0618 | 0.028 | 0.309 |
|  |  | 5e-4 | 0.0553 | 0.0354 | 0.106 | 0.0801 | 0.0357 | 0.319 |
|  |  | 5e-2 | 0.0127 | 0.00889 | 0.0351 | 0.672 | 0.321 | 2.87 |

Table 5: $m = 1000$, $n = 1000$

| Method | Reg. Ent | Reg. GL | Runtime (s) ↓ Median | Runtime (s) ↓ $q10$ | Runtime (s) ↓ $q90$ | Agg. $W_2$ dist. ↓ Median | Agg. $W_2$ dist. ↓ $q10$ | Agg. $W_2$ dist. ↓ $q90$ |
|---|---|---|---|---|---|---|---|---|
| GLSK | 1e-3 | 1e-6 | 4.97 | 4.91 | 9.27 | 1.66 | 0.405 | 11 |
|  | 1e-3 | 5e-4 | 4.9 | 4.82 | 9.19 | 1.66 | 0.405 | 11 |
|  | 1e-3 | 5e-2 | 5.52 | 4.91 | 15.9 | 1.64 | 0.389 | 11 |
|  | 1e-2 | 1e-6 | 4.9 | 4.79 | 5.53 | 3.06 | 0.926 | 8.97 |
|  | 1e-2 | 5e-4 | 4.79 | 4.66 | 5.5 | 3.06 | 0.926 | 8.97 |
|  | 1e-2 | 5e-2 | 4.88 | 4.73 | 5.55 | 3.15 | 0.926 | 9.03 |
|  | 1e-1 | 1e-6 | 4.71 | 4.68 | 4.73 | 20.6 | 8.97 | 74.3 |
|  | 1e-1 | 5e-4 | 4.74 | 4.7 | 4.78 | 20.6 | 8.97 | 74.3 |
|  | 1e-1 | 5e-2 | 4.76 | 4.73 | 4.82 | 20.6 | 8.97 | 74.3 |
|  | 1. | 1e-6 | 4.61 | 4.57 | 4.64 | 130 | 37.5 | 403 |
|  | 1. | 5e-4 | 4.62 | 4.6 | 4.67 | 130 | 37.5 | 403 |
|  | 1. | 5e-2 | 4.74 | 4.71 | 4.77 | 130 | 37.5 | 403 |
| GLDROT |  | 1e-6 | 0.109 | 0.0735 | 0.146 | 0.0999 | 0.0424 | 0.304 |
|  |  | 5e-4 | 0.0598 | 0.045 | 0.0763 | 0.138 | 0.0632 | 0.44 |
|  |  | 5e-2 | 0.0281 | 0.0232 | 0.0374 | 3.16 | 1.14 | 9.01 |

Table 6: $m = 1000$, $n = 1500$

| Method | Reg. | | Runtime (s) ↓ | | | Agg. $W_2$ dist. ↓ | | |
|---|---|---|---|---|---|---|---|---|
| | Ent | GL | Median | $q10$ | $q90$ | Median | $q10$ | $q90$ |
| GLSK | 1e-3 | 1e-6 | 3.82 | 3.75 | 9.03 | 0.311 | 0.0657 | 6.48 |
| | 1e-3 | 5e-4 | 3.8 | 3.75 | 9 | 0.311 | 0.0657 | 6.48 |
| | 1e-3 | 5e-2 | 3.8 | 3.77 | 10.6 | 0.345 | 0.0665 | 6.48 |
| | 1e-2 | 1e-6 | 3.78 | 3.73 | 6.8 | 1.21 | 0.69 | 5.47 |
| | 1e-2 | 5e-4 | 3.8 | 3.76 | 6.82 | 1.21 | 0.69 | 5.47 |
| | 1e-2 | 5e-2 | 3.82 | 3.79 | 6.88 | 1.21 | 0.69 | 5.47 |
| | 1e-1 | 1e-6 | 3.7 | 3.67 | 3.72 | 8.24 | 3.47 | 31.6 |
| | 1e-1 | 5e-4 | 3.75 | 3.73 | 3.78 | 8.24 | 3.47 | 31.6 |
| | 1e-1 | 5e-2 | 3.77 | 3.73 | 3.81 | 8.24 | 3.47 | 31.6 |
| | 1. | 1e-6 | 3.73 | 1.86 | 3.77 | 45.5 | 9.92 | 218 |
| | 1. | 5e-4 | 3.69 | 1.85 | 3.73 | 45.5 | 9.92 | 218 |
| | 1. | 5e-2 | 3.75 | 1.88 | 3.78 | 45.5 | 9.92 | 218 |
| GLDROT | | 1e-6 | 0.113 | 0.0758 | 0.147 | 0.0475 | 0.0215 | 0.137 |
| | | 5e-4 | 0.0745 | 0.0549 | 0.0951 | 0.0529 | 0.0245 | 0.163 |
| | | 5e-2 | 0.0232 | 0.0178 | 0.0288 | 0.331 | 0.154 | 1.38 |

# G  Model specification of the generative Adversarial Model

We adopted similar network structures and used the same loss function as in [35] and performed several experiments with image generation based on the MNIST and CIFAR10 datasets.

To define the loss, we need the following sample-based OT-cost.

$$\mathcal{W}_{c,h}(\mathbf{X}, \mathbf{Y}) = \mathrm{OT}_h(C_{\mathbf{X},\mathbf{Y}})$$

$$:= \inf_{M \in \mathbb{R}_+^{m \times n}} \left\{ \langle C_{\mathbf{X},\mathbf{Y}}, M \rangle + h(M) : M\mathbf{1}_n = m^{-1}\mathbf{1}_m, M^\top \mathbf{1}_m = n^{-1}\mathbf{1}_n \right\}.$$

Here $m$ and $n$ are the batch sizes corresponding to $\mathbf{X}$ and $\mathbf{Y}$, and $C_{\mathbf{X},\mathbf{Y}}$ is a matrix with pairwise distances between the samples of $\mathbf{X}$ and $\mathbf{Y}$. The mini-batch energy distance [35] is then given by

$$\mathcal{L}_h = \mathcal{W}_{c,h}(\mathbf{X}, \mathbf{Y}) + \mathcal{W}_{c,h}(\mathbf{X}', \mathbf{Y}) + \mathcal{W}_{c,h}(\mathbf{X}, \mathbf{Y}') + \mathcal{W}_{c,h}(\mathbf{X}', \mathbf{Y}')$$
$$- 2\mathcal{W}_{c,h}(\mathbf{X}, \mathbf{X}') - 2\mathcal{W}_{c,h}(\mathbf{Y}, \mathbf{Y}').$$

Here, $\mathbf{X}$, $\mathbf{X}'$ are two independent mini-batches from real data, while $\mathbf{Y}$, $\mathbf{Y}'$ are two independent mini-batches generated from the generator.

In the experiments, we used the cosine similarity to parameterize the cost, and we replaced the OT solver used in [35] - the Sinkhorn-Knopp algorithm, by the PyTorch wrapper for our OT solver with the quadratic regularizer.

The model structures are adapted from the ones in [35]. Weight normalization is used to construct the parameters of the models [34], and the activation functions are gated linear units [13] and concatenated ReLUs [38]. We train the model with the Adam optimizier [22] using an initial learning rate of $3 \times 10^{-4}$, with $\beta_1 = 0.5$ and $\beta_2 = 0.999$. The batch size is 1024 for the MNIST experiment and 2048 for the CIFAR10 experiment. We update the generator three times for every discriminator update. As for the parameters of our OT solver, we use $\epsilon = 10^{-4}$ as the stopping criterion and $\lambda = 10^{-3}$ as the weight of the quadratic regularizer. The results for MNIST and CIFAR10 are detailed below.

Different combinations of hyperparameters, model structures and schedules for the weight of the regularizers and the stopping criterion (the OT solver can allowed to produce low-accuracy solutions at the beginning of the training, for example) may potentially improve the results of the generated samples, but since GANs are not the focus of this paper, we leave such refinements as future work.

### G.1   MNIST Architecture and Results

| Operation | Activation | Kernel Size | Stride | Padding | Output Shape |
|---|---|---|---|---|---|
| Sample $z$ | | | | | $[4]$ |
| Linear | GLU | | | | $[128 \cdot 7 \cdot 7]$ |
| Reshape | | | | | $[128, 7, 7]$ |
| Upsample $\times 2$ | | | | | $[128, 14, 14]$ |
| 2D Convolution | GLU | $[5, 5]$ | 1 | same | $[128, 14, 14]$ |
| Upsample $\times 2$ | | | | | $[128, 28, 28]$ |
| 2D Convolution | GLU | $[5, 5]$ | 1 | same | $[64, 28, 28]$ |
| 2D Convolution | tanh | $[5, 5]$ | 1 | same | $[1, 28, 28]$ |

Table 7: Generator Architecture for MNIST

| Opertion | Activation | Kernel Size | Stride | Padding | Output Shape |
|---|---|---|---|---|---|
| 2D Convolution | CReLU | $[5, 5]$ | 1 | same | $[128, 28, 28]$ |
| 2D Convolution | CReLU | $[5, 5]$ | 1 | same | $[128, 28, 28]$ |
| 2D Convolution | CReLU | $[5, 5]$ | 2 | 2 | $[256, 14, 14]$ |
| 2D Convolution | CReLU | $[5, 5]$ | 2 | 2 | $[256, 7, 7]$ |
| Flatten | | | | | $[256 \cdot 7 \cdot 7]$ |
| L2 Normalization | | | | | $[256 \cdot 7 \cdot 7]$ |

Table 8: Discriminator Architecture for MNIST

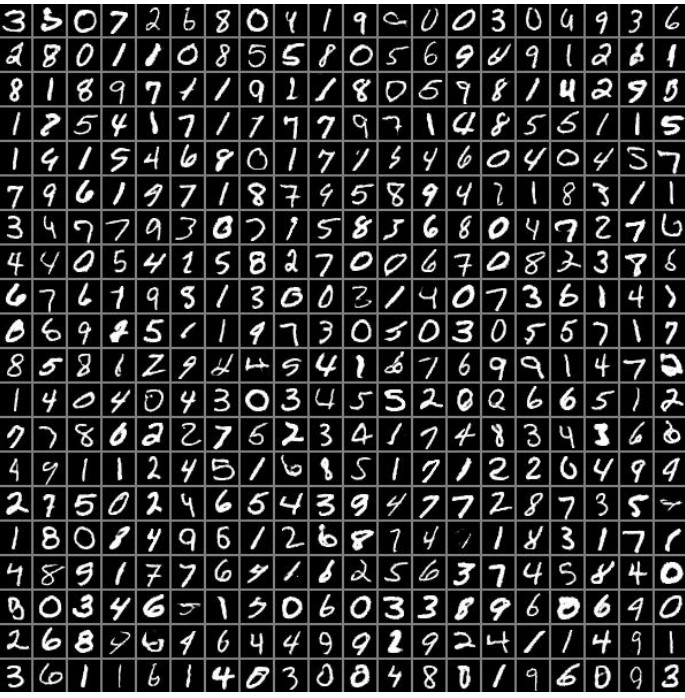

Figure 8: Generated samples at 100th epoch for MNIST (Original size for each sample is 28x28 px)

The generated samples in Figure 8 are clear and shape, and no mode collapse can be observed (digits from 0 to 9 with various variants can be found in the figure). Compared with the results in [17] (Figure 3), our results show higher quality.

## G.2 CIFAR10 Architecture and Results

| Opertion | Activation | Kernel Size | Stride | Padding | Output Shape |
|---|---|---|---|---|---|
| Sample $z$ | | | | | $[100]$ |
| Linear | GLU | | | | $[1024 \cdot 4 \cdot 4]$ |
| Reshape | | | | | $[1024, 4, 4]$ |
| Upsample $\times 2$ | | | | | $[1024, 8, 8]$ |
| 2D Convolution | GLU | $[5, 5]$ | 1 | same | $[512, 8, 8]$ |
| Upsample $\times 2$ | | | | | $[512, 16, 16]$ |
| 2D Convolution | GLU | $[5, 5]$ | 1 | same | $[256, 16, 16]$ |
| Upsample $\times 2$ | | | | | $[256, 32, 32]$ |
| 2D Convolution | GLU | $[5, 5]$ | 1 | same | $[128, 32, 32]$ |
| 2D Convolution | tanh | $[5, 5]$ | 1 | same | $[3, 32, 32]$ |

Table 9: Generator Architecture for CIFAR10

| Opertion | Activation | Kernel Size | Stride | Padding | Output Shape |
|---|---|---|---|---|---|
| 2D Convolution | CReLU | $[5, 5]$ | 1 | same | $[256, 32, 32]$ |
| 2D Convolution | CReLU | $[5, 5]$ | 2 | 2 | $[512, 16, 16]$ |
| 2D Convolution | CReLU | $[5, 5]$ | 2 | 2 | $[1024, 8, 8]$ |
| 2D Convolution | CReLU | $[5, 5]$ | 2 | 2 | $[2048, 4, 4]$ |
| Flatten | | | | | $[2048 \cdot 4 \cdot 4]$ |
| L2 Normalization | | | | | $[2048 \cdot 4 \cdot 4]$ |

Table 10: Discriminator Architecture for CIFAR10

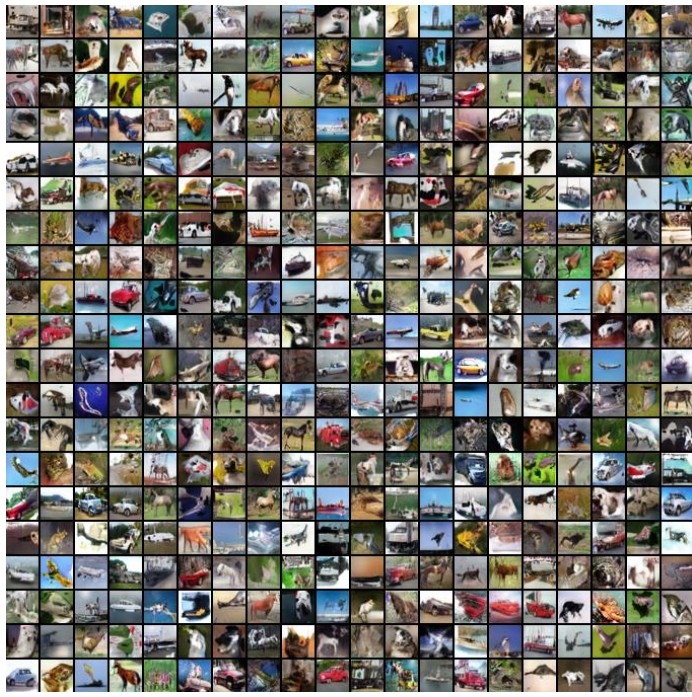

Figure 9: Generated samples at 1900th epoch for CIFAR10 (Original size for each sample is 32x32 px)

In Figure 9, most samples are recognizable, and the samples cover the 10 categories of the CIFAR10 dataset with enough diversity. Our generated samples have similar quality as the ones reported in [35] (Figure 4) which uses the Sinkhorn-Knopp algorithm as the OT solver in the mini-batch energy loss.

