# OpenReview forum: "Bringing regularized optimal transport to lightspeed: a splitting method adapted for GPUs"
_NeurIPS.cc/2023/Conference — NeurIPS 2023 poster_

### Official Review · Reviewer_4HzB · 2023-06-11

**Soundness:** 3 good
**Presentation:** 4 excellent
**Contribution:** 3 good
**Rating:** 7
**Confidence:** 4

**Summary:**

This paper adapts the Douglas-Rachford splitting to solve a wide range of sparsely-regularized optimal transport problems efficiently using GPU-parallelizable operations. The contributions are as follows:

1) Adapt the Douglas-Rachford splitting to handle regularized OT problems with sparsity-inducing penalties
2) Prove global convergence of the method and prove an accelerated local linear rate once the support is identified
3) Provide GPU kernels for well-known regularizers (quadratic, group lasso) that achieve low-cost per iteration



**Strengths:**

The paper is well-written, the literature review is exhaustive and sets the context for this work: the relevant references are correctly cited. Besides, the subject tackled by the paper is critical for a wider adoption of OT to large-scale problems. The supplementary materials do a great job at making the experiments reproducible and explaining the low-level details of the kernel implementation.

**Weaknesses:**

The area of improvement for this paper remains the experiments section. For benchmarking purposes, I would use the Benchopt tool: https://github.com/benchopt/benchopt. Benchopt offers reproducible benchmarks and convergence curves over multiple runs. A figure from Benchopt to compare RDROT to L-BFGS would be more impactful than figure 2.

Moreoever, the authors should favour real-world datasets instead of simulated cost matrices. The quadratic regularization and the Group Lasso sections should both display examples on real-world datasets to prove the robustness of the proposed method.

=========

Besides, I list below a few typos I spotted in the paper:

1) l.49: Douglas-Rachford... splitting is missing.

2) l.93: $\lvert \lvert X \rvert \rvert_F = \sqrt{\langle X, X \rangle}$.

3) l.108: I would precise that $\lambda > 0$.

4) In equation 6, $f$ and $e$ are extracted from [23], without explicitly defining them. I would explicitly use $\mathbb{1}_n$ for $e$ and $\mathbb{1}_m$ for $f$.

5) l.165: "used" twice.

**Questions:**

l.143: "since $Y_k$ can be eliminated...": could you elaborate on what you mean? After multiple readings, this remains unclear.

**Limitations:**

No negative societal impact for this work.

---

> ### Author Rebuttal · Authors · 2023-08-09
>
> Thank you for the time and effort you invested in thoroughly reviewing our paper. Your feedback is very valuable to us, as it helps us improve our paper. We are grateful that you highlighted some typos in the manuscript -  we will make sure to revise it according to your comments in upcoming versions.
>
> Regarding your question on why $Y_k$ can be eliminated, this comes from the fact that $Y_k$ can be substituted in the $X$-update of equation (5). This results in that $X_{k+1}$ only depends on $X_k$, $\phi_k$, and $\varphi_k$. Further, since the $\phi$-updates are only dependent on $X$, we don’t have to keep track of $Y$ anymore after doing this substitution. This is pivotal for the efficiency of the resulting algorithm since it enables us to only keep track of one large matrix instead of two (or even three). To improve the readability of the final manuscript, we will add some additional explanations here.
>
> Unfortunately, up to this day, there are no OT benchmarks included in BenchOpt that we can use. Of course, this also means that there is an opportunity for us to contribute to the BenchOpt effort. Specifically, it would’ve been nice to have customized a QuadOT benchmark and GL-OT benchmark for this paper, but due to time constraints of the rebuttal, we haven’t been able to prioritize this. We will look into adding BenchOpt experiments to our main repository, to make our results more convincing and reproducible, and reach out to the BenchOpt authors to investigate the possibility to have these benchmark problems merged with their efforts.
>
> We appreciate you sharing your input on improvements to the paper. We will update the manuscript according to the typos you highlighted in your review.
>
> Many thanks!
>
>
> The authors

---

> > ### Comment · Reviewer_4HzB · 2023-08-16
> >
> > I'd like to thank the authors for their answer.
> > This is well noted for the BenchOpt benchmark.
> >
> > Keeping my grade unchanged as it is already positive.

---

### Official Review · Reviewer_PMSB · 2023-06-29

**Soundness:** 2 fair
**Presentation:** 3 good
**Contribution:** 3 good
**Rating:** 7
**Confidence:** 3

**Summary:**

The paper develops an approximation algorithm for solving the optimal transport (OT) problem with a general class of regularizers (named "sparsity promoting", but more precisely characterized by not penalizing sparse solutions). The Douglas-Rachford splitting algorithm is used, extending the  previously introduced DROT algorithm to handle regularization. After introducing the OT problem and the Douglas-Rachford algorithm, the authors define a class of regularizers which include quadratic and group lasso regularization.
With additional assumptions, the convergence rate of the algorithm is established via first a convergence to the correct support, and then linear convergence to the optimal solution. The authors state that the proposed algorithm converges to epsilon accuracy in 1/epsilon iterations, instead of 1/(epsilon^2) of previously known methods. A fast GPU implementation of the RDROT algorithm is briefly described, along with some other practical considerations. Two numerical experiments are performed on a) domain adaptation, and b) generative modeling.

**Strengths:**

The paper is well written overall, with a clear introduction to the problem, proposed approach and contributions. The practical considerations are also well written, and the GPU implementation -- as well as integration in popular programming frameworks -- makes the contribution particularly useful to the community.
The experiments are clear and present a convincing case for RDROT.

**Weaknesses:**

The theoretical aspects could be made clearer: it is not evident how theorems 1 and 2 lead to a 1/epsilon convergence rate (as there is no rate for support identification). The relevance of assumption 1 to the OT problem should be expanded upon: if and where it has been used in the literature, and when it doesn't apply.
In particular, a comparison to other convergence guarantees existing in the literature for Sinkhorn divergences (e.g. [1]) would be very useful in assessing the applicability of the proposed method.

[1] Near-linear time approximation algorithms for optimal transport via Sinkhorn iteration, Jason Altschuler, Jonathan Weed, Philippe Rigollet, 2017

**Questions:**

If at all feasible, a comparison to more Sinkhorn-based algorithms (also in simpler settings, where convergence is assessed) could be useful to place the work within the literature.

**Limitations:**

A discussion on the tradeoffs between the proposed algorithm and minimizing sinkhorn divergences would certainly help to correctly evaluate the proposed contribution.

---

> ### Author Rebuttal · Authors · 2023-08-09
>
> Thank you for your questions and your feedback on our paper! We believe that we have been able to address all your concerns, as detailed below.
>
> We agree that there is room for improvement to make the theory section clearer. The 1/epsilon convergence rate of DR-splitting is nothing that we derive, but we simply refer to the work by He et al (2012) . This is a general result for DR splitting that is derived by leveraging the firm non-expansiveness of the proximal operators. The result only demands that the OT problem is convex and closed and that the problem has a solution. Hence, the 1/k convergence follows for all closed and convex regularizers that are proper over the convex polytope. Further, although the results in He et al (2012) are for ergodic sequences, some of the results can be generalized to non-ergodic sequences, see, e.g., He (2015). We will clarify this in the final version of our document.
>
> In addition to such existing results, we establish that our algorithm identifies the correct sparsity structure in a finite number of iterations and that stronger (typically linear) rates dominate once the correct sparsity structure is identified. This means that we not only have a global $1/k$ rate that is competitive to many alternative algorithms, but that we also, in many cases, have considerably stronger local rates. These stronger results are established under the additional Assumption 1. As discussed in Section A.2 in the Supplementary material, Assumption 1 holds for most OT problems besides some very specific edge cases. We have never encountered such edge cases in practice, and believe that they are rare. You are right that this assumption is typically not imposed in other OT papers. However, in the context of computational OT, to the best of our knowledge, no other work leverages sparsity to derive theoretical guarantees. Papers in other areas that do, such as Liang et al. (2015) , typically make this or related assumptions.  We will clarify these points in the final version of the document.
>
> The reference you mention in the review (Altchuler et al. (2017)), but also Lin et al (2019), which we refer to in the paper, derive $1/\epsilon^2$ rates for Sinkhorn and Greenkhorn. These rates are significantly worse than the ergodic rates that hold for RDROT. However, an interesting aspect of their analysis is that they quantify the total variation errors (i.e. $\ell_1$-norm) of the marginals in terms of the problem size. We have not (yet) attempted to derive such results. Nevertheless, our numerical results strongly suggest that the numerical advantages of RDROT do not diminish as the problem size is increased (at least up to problems of the sizes that fit the memory of our GPU card).
>
> We hope our rebuttal will make you more confident that the community would benefit from learning about our contributions at the NeurIPS Conference 2023.
>
> Thank you!
>
> The authors
>
> **References**
>
>
> He B, Yuan X. On the O(1/n) convergence rate of the Douglas–Rachford alternating direction method. SIAM Journal on Numerical Analysis. 2012
>
> He B, Yuan X. On non-ergodic convergence rate of Douglas–Rachford alternating direction method of multipliers. Numerische Mathematik. 2015.
>
> Liang J, Fadili J, Peyré G, Luke R. Activity identification and local linear convergence of Douglas–Rachford/ADMM under partial smoothness. InScale Space and Variational Methods in Computer Vision: 5th International Conference. 2015.
>
> Altschuler J, Weed J, Rigollet P, Near-linear time approximation algorithms for optimal transport via Sinkhorn iteration. 31st Conference on Neural Information Processing Systems, 2017.
>
> Tianyi Lin, Nhat Ho, and Michael Jordan. On efficient optimal transport: An analysis of greedy and accelerated mirror descent algorithms.  International Conference on Machine Learning, 2019.

---

> > ### Comment · Reviewer_PMSB · 2023-08-13
> >
> > Having read the other reviews and the author's rebuttal, I am more confident of the strength of this submission. In particular I believe that while the experimental section may not be perfect, it is still worthy of publishing a work which improves efficiency by technical (i.e. implementation) means as well as algorithmically. I appreciate the effort of the authors in improving the clarity of the theory section.

---

> > > ### Author Response · Authors · 2023-08-16
> > >
> > > Dear Reviewer PSMB,
> > >
> > > Thank you for reading and reflecting on the other reviews and our rebuttal. We are very pleased to learn that you have increased your initial score - thank you!
> > >
> > > We are currently exploring a number of additional examples that we hope to add to our code repository (a reference to the repo will be added to the paper upon acceptance), and are looking into how we could leverage the BenchOpt infrastructure to simplify reproducibility of the experiments and comparison with alternatives. We hope that this will constitute a good complement to the experimental results already included in the paper.
> > >
> > > Sincerely,
> > >
> > > The Authors

---

### Official Review · Reviewer_e6yJ · 2023-07-03

**Soundness:** 4 excellent
**Presentation:** 4 excellent
**Contribution:** 3 good
**Rating:** 7
**Confidence:** 3

**Summary:**

The paper presents an extension of the Douglas-Rachford algorithm in [23] to regularised optimal transport. It describes conditions under which sparsity-inducing regularisations result in sparse solutions and convergence rates of the resulting estimates.
Implementation on GPU, as well as gradient routines, are considered in detail. Some numerical experiments illustrate the performance of the method.

**Strengths:**

To the best of my knowledge, the introduction of convex regularisation as part of a DR algorithm is novel. It seems to bring several benefits, both computational (speed + stability) and theoretical (sparsity and convergence rate guarantees).

The article is comprehensive in its treatment, considering parallelisation, gradient computation, and algorithmic designs such as step-size selection, etc.

While I have not had time to go through the supplementary material in detail (I hope to be able to do so later on), the theoretical analysis looks sound and reasonable.

**Weaknesses:**

The experimental comparison with [4] is unfair: the implementation of the proposed method is fully tailored,  while the choice for [4] is a generic and not optimised PyTorch implementation. To improve this, the same effort should be spent on developing a custom CUDA kernel for [4] as for RDROT or providing a generic PyTorch implementation for both, rather than optimising the proposed method only. **IF** it is impossible that [4] can be improved, this must be explicitly discussed and explained.

Additionally, no comparison with Sinkhorn for entropy-regularised OT (and its accelerated versions, see for example http://angkor.univ-mlv.fr/~vialard/wwwsrc/AndersonMultiMarginal.html and https://github.com/ott-jax/ott/blob/main/src/ott/solvers/linear/acceleration.py)
is given. Arguably, this is not sparsity-inducing, but (i) the title and introduction directly refer to [10], and (ii) the smoothness of the transport plan is sometimes a desirable property. Again, the comparison should be done with the same amount of implementation effort (either both custom-made kernels, or both high-level).

While Section 3.3 discusses gradient computation, the supplementary is limited to gradient w.r.t. the cost function. In general, people tend to also be, if not more, interested in the gradient w.r.t. the marginals (weights and positions). Some discussion on this would be welcome. I also believe that (for at least the quadratic regularisation) the optimal transport plan is not differentiable w.r.t. the OT inputs. A quick discussion of this in the supplementary would be welcome.

Finally, in terms of literature review, I am surprised that https://epubs.siam.org/doi/10.1137/130920058 (published in 2014) is not mentioned given that all the methods introduced in [23] (published in 2022) can, as far as I understand, be found in it. This does not impact the novelty of the current work (as I am not aware of a regularised version thereof, but proper attribution of the DR algorithm should go tohttps://epubs.siam.org/doi/10.1137/130920058 rather than [23].



**Questions:**

My main point of concern is the unfair empirical comparison. Because of this, I cannot trust the paper's title claim of "lightspeed", given that its performance may very well be implementation specific rather than methodological. I would be willing to substantially strengthen my rating if this was solved. See also the other (more minor and easy to fix) weaknesses.


#### Minor points:
- The notation for the prox operator $\mathrm{prox}_{\rho f}$ is undefined in the paper.
- $e$ and $f$ in Equation (6) are undefined. Please replace with $\mathbf{1}_{m/n}$ as necessary.
- Typos: I did not spot any in the main text, but spotted two in the supplementary (assumpion, opertion), there may be other ones.

---

> ### Author Rebuttal · Authors · 2023-08-09
>
> We are grateful for the time you invested in reviewing our paper. Your feedback means a lot to us - thank you!
>
> We have rerun our experiments with a PyTorch version of RDROT. Naturally, this results in a performance drop for our algorithm, but the experiments still show that our algorithm is faster than the state-of-the-art. Please see Figures 1 and Figure 2 in the authors' rebuttal. With that said, since the main computational bottleneck in RDROT is the update $X \leftarrow X -\rho C$, it is difficult to fully utilize the GPU parallelization and efficient memory management without tailoring a kernel (that is, by using working blocks and warp reductions). Sinkhorn-based methods are different since the memory-intensive operations are matrix-vector multiplications, which libraries such as PyTorch and TensorFlow handle efficiently. If you think commenting on this will improve the paper, we add a short comment in the final version of the paper.
>
> “Lightspeed computations” in this context refers to Cuturi’s seminal paper on Sinkhorn’s algorithm for OT. Therefore, when we assert that we bring regularized OT to lightspeed, we mean that we manage to develop an algorithm for regularized OT that has similar, or better performance than Sinkhorn (on the entropically regularized OT problem). As Figure 2 in the Supplementary material, and Figure 2 in the PDF attached to the Author rebuttal suggest, the per iteration cost of our method is similar to that of Sinkhorn. This together with the improved iteration complexity, in our view, suffice to say that our approach estimate transportation plans with lightspeed computations. If you believe that this would improve our paper, then we could also include explicit figures of the wall-clock execution times of the two methods.
>
> With this understanding, techniques that accelerate the practical converge of SK execute “faster than light speed”. Our focus in this paper is really on regularized OT problems, and not on the pure OT problem that Mai et al considered, nor on multimarginal problems that is done in the post that you shared. For regularized problems, we are already able to attain significant speed-ups relative to the state of the art. It is an interesting idea to apply Anderson Acceleration also to the RDROT iterations, but it would be memory intense and probably difficult to scale even to medium-size OT problems.
>
> To differentiate with respect to the marginals, one can use a similar argument discussed in the paper, but on the dual problem. We elaborate on this in the Author's Rebuttal. To our knowledge, most sparsity-promoting regularizers will result in transportation plans that are differentiable at least almost everywhere - but the resulting Jacobians will be zero. Developing ways of working around this is an active area of research, e.g. Sahoo et al. 2023, and it would be interesting to explore how this further can be applied to this framework.
>
> We appreciate you sharing the typos you’ve found in our manuscript. We will revise according to your comments when polishing the final version.
>
> Thanks a lot!
>
> The authors
>
> **References**
>
> Sahoo SS, Paulus A, Vlastelica M, Musil V, Kuleshov V, Martius G. Backpropagation through combinatorial algorithms: Identity with projection works, International Conference on Learning Representations 2023

---

> > ### Comment · Reviewer_e6yJ · 2023-08-14
> > **Acknowledgement of the rebuttal**
> >
> > I thank the authors for this very clear and detailed rebuttal. I will, as I planned, raise my score, on grounds of improved soundness. However, this point
> >
> > > Finally, in terms of literature review, I am surprised that https://epubs.siam.org/doi/10.1137/130920058 (published in 2014) is not mentioned given that all the methods introduced in [23] (published in 2022) can, as far as I understand, be found in it
> >
> > was not addressed by the reviewers. I would like to insist that the original attribution of the DROT methodology should go to it rather than [23].
> >
> > On a side note, I have not found time to thoroughly check the proof for the sparsity-inducing properties, but have skimmed through it once more and still found it sound at a superficial level. I however can't raise my confidence score because of this.

---

> > > ### Author Response · Authors · 2023-08-15
> > >
> > > Dear Reviewer e6yJ,
> > >
> > > we are pleased to learn that you are satisfied with our rebuttal and increased your score to "Accept". We apologize that our rebuttal forgot to discuss the missing citations in our first draft. We will (and always intended to) add them, in particular https://epubs.siam.org/doi/10.1137/130920058, to the final version of the paper.
> > >
> > > Sincerely,
> > > The Authors

---

### Official Review · Reviewer_WrJb · 2023-07-13

**Soundness:** 3 good
**Presentation:** 3 good
**Contribution:** 2 fair
**Rating:** 4
**Confidence:** 4

**Summary:**

In previous work [23] (DROT), the Douglas-Rachford splitting was applied to solving unregularized optimal transport (OT). The idea is to split the original variable into two variables $X$ and $Z$, where $X\ge 0$ and $Z$ prescribed row and column sums. The current paper extends this idea to regularized OT. Compared to previous work [23], there are some novelties such as the local linear rate of convergence (Section 3.1, based on the results of Liang et al. [19]) and differentiation (Section 3.3).

**Strengths:**

The paper is well written in general. Compared to DROT [23], this paper provides a local linear rate of convergence and a differentiation result. I particularly like the latter, it's a nice result and is useful in deep learning applications.
The obtained algorithm with its GPU implementation is very efficient and has the potential to replace existing algorithms in many applications (though I have to say that this is rather an encouragement, because the same thing could have been said about DROT [23], yet after more than two years since its publication, it hasn't been adopted in any research paper (3 citations at the time of writing, including one from the current submission), I wonder why this is the case.

**Weaknesses:**

The main limitation of this paper lies in its algorithmic contributions: there is virtually none. Unfortunately this is a major issue. The extension of DROT to the regularized case is rather straightforward. This is a very nice extension of DROT, but it alone is a rather weak contribution.


Some presentation issues:

- You should cite Sinkhorn and Knopp [31] when first mentioning the Sinkhorn algorithm in the introduction. Curuti's work [20] should be cited only for its efficient implementation.

- You should cite Bauschke et al. 2021 (Projecting onto rectangular matrices with prescribed row and column sums) at line 138 when mentioning the projection onto X. It seems to me that this projection is the heart of both DROT and your algorithm.

- Regarding the definition 2.1 of "Sparsity promoting regularizers" on page 3: according to this definition, it seems that the entropic regularizer is also a sparsity promoting one. However, the current discussion seems to indicate that the entropic regularizer is not suited for sparsity. Thus more clarifications and discussions are needed here (or maybe just choose another name for the definition).

- I think the details on the stopping conditions and backpropagation are important and should be presented in the main content rather than in the appendix.

Minor typos at line 93: <X,X> instead o <X,Y> in the definition of the norm.

**Questions:**

What happens if we use the entropic regularizer in RDROT?

**Limitations:**

See weaknesses.

---

> ### Author Rebuttal · Authors · 2023-08-09
>
> Thank you for your thoughtful review and constructive feedback. We are grateful for your positive words about our work but, quite naturally, disagree with the statement that the algorithmic contributions are limited, or that this should be reason enough to reject the paper. Let us elaborate.
>
> We believe that an important message in the work by Mai et al. 2022 was to highlight that for most existing algorithms, the unregularized OT problem is memory bound. In other words, even though there are many algorithms for solving the OT class of linear programs with an excellent iteration complexity, they are not amendable to efficient implementations beyond toy examples. In the same spirit, it is our opinion that the main contribution by Mai et al. was not to simply apply DR splitting to OT but rather to do it in a way that enables a memory-efficient GPU implementation.
>
> A limitation of the work by Mai et al. 2022 is that they focused on the unregularized OT problem. Although they reported crisp transport plans with significantly reduced blur compared to SK solutions, such high-accuracy solutions may not be needed in scenarios where the underlying data is noisy. This limits the practical usefulness of the DROT algorithm.
> The main contribution of this work has been to identify a broad class of regularizers that can be dealt with efficiently on GPUs. Such problems have many applications but, quite surprisingly, few (if any) efficient algorithms. On the contrary, the algorithm that we propose is surprisingly simple, yet amazingly efficient. We believe that there are very few NeurIPS contributions that can, as we do, present an algorithm that runs 100x faster than the state-of-the-art on a well-established problem class. Indeed, one may argue that such an achievement alone should make the paper very interesting to a significant part of our community, and therefore qualify it for publication. To the best of our knowledge, the class of sparsity promoting regularizers is novel in the context of optimal transport, and its usefulness has not been pointed out in the literature before.
>
> As you, and the other reviewers, point out, these are not the only contributions of the paper. We also do provide global and local theoretical guarantees that apply to the sparsity-promoting regularizers. Further, we show in our experiments that our framework is readily applicable to a range of problems, including domain adaptation and learning of generative models. To facilitate for the practitioner, we also wrapped RDROT in the autograd frameworks PyTorch and TensorFlow.
>
> We believe there has been a minor misunderstanding regarding the question on entropic regularization. Entropic regularization: $H(X) = \sum_{ij} h(X_{ij})$ with $h(x) = x \log x$, is not sparsity-promoting, since it is not closed. By extending its domain so that$h(0) = 0$, it becomes closed, but still not sparsity promoting as $h(1/e) = -1/e < h(0)$. Therefore, our framework, in its current form, cannot handle this particular regularizer. However, if entropy-regularized OT is of interest, Sinkhorn-based approaches do an excellent job estimating optimal transportation plans. What we are trying to accomplish in our work is to go beyond this regularization scheme. One interesting direction to explore further is using hyperentropic regularization to interpolate between quadratic regularization and entropic regularization, which can be handled by our method.
>
> Further, we think your input on the presentation issues are fair -  we will add the citations you recommended and fix the typo. We considered your suggestion to move the stopping criterion from the appendix, but we found that the current organization of the manuscript gives a better flow of ideas, and is more readable.
>
> We hope that our answers will make you reconsider your scores.
>
> Thanks!
> The authors
>
> **References**
>
> Vien V Mai, Jacob Lindbäck, and Mikael Johansson. A fast and accurate splitting method
> for optimal transport: Analysis and implementation. International Conference on Learning
> Representations, 2022

---

> ### Comment · Area_Chair_hGLj · 2023-08-18
>
> Dear WrJb: can you read the authors' response, and see if your comments are addressed?

---

### Author Rebuttal · Authors · 2023-08-09

Dear AC and reviewers,

Thank you for the time you invested in the peer-reviewing process. The input you provided has both been insightful and inspirational for us.

The reviewers have recognized many strengths and novelties of our contributions, including:

- the generality of the algorithm for regularized OT (it is readily applicable to many regularization schemes and constraints). We achieve this without compromising the numerical performance or the theoretical guarantees (e6yJ, PMSB, 4HzB).
- by using a tailored GPU kernel, our algorithm leads to ~100 x speedup compared to the state-of-the-art for several problems (WrJb, e6yJ, PMSB, 4HzB).
- we establish that a global competitive 1/k rate that holds for all considered regularizers, and a local linear rate under additional, weak assumptions (WrJb, e6yJ, PMSB).

For clarity, when we state that we bring regularized OT to lightspeed, what we mean is that RDROT has computational advantages comparable with Cuturi’s Sinkhorn method. We provide theoretical, numerical, and computational justifications for this claim. Our method enjoys rates that are better than that of the state-of-the-art, its per-iteration cost is low, and it benefits from GPU parallelization. Since our algorithm depends on operations that are difficult to parallelize in high-level frameworks such as PyTorch, to fully utilize the computational potential of the GPU, we’ve developed a GPU kernel for our framework. During this rebuttal, however, we found that an RDROT version implemented in PyTorch is still competitive for many problems. We illustrate this in Figure 1 in the attached PDF, in which we compare two versions of QDROT (RDROT applied to quadratically regularized OT), to an LBFGS method. Although the torch version is almost 10 times slower, it is still significantly faster than the LBFGS method, which is among the more popular algorithms for this application.

To enable practitioners to use our frameworks for deep learning, we developed a PyTorch and a TensorFlow wrapper that feature automatic differentiation through regularized OT costs. Some reviewers showed interest in this, including WrJb and e6yJ, which we are very pleased about. However, we stress that there are many possible extensions here. For instance, to differentiate with respect to the marginals, one can use the estimated dual variables $\phi/\rho$, and $\varphi/\rho$ as gradient approximations (see PDF for more details). To differentiate with respect to the transportation plan itself, one must use different techniques since the derivative is typically zero almost everywhere. We believe that the techniques proposed by Sahoo et al. (2023) could be worth exploring in this framework.

The feedback provided by the reviewers will contribute towards an even stronger final manuscript. The questions and criticisms raised by the reviewers have been taken into careful consideration - we are confident that we addressed all major concerns and the vast majority of the more minor suggestions. We believe our paper - which has been improved further by this reviewing process - would be of great value for the community, and hope that you agree that it is worthy of being presented at the conference in December.

Thank you!


The authors

**References**
Sahoo SS, Paulus A, Vlastelica M, Musil V, Kuleshov V, Martius G. Backpropagation through combinatorial algorithms: Identity with projection works, International Conference on Learning Representations 2023

---

### Decision · Program_Chairs · 2023-09-21

**Decision:**

Accept (poster)

**Comment:**

This paper extends the Douglas-Rachford (DR) splitting method to sparsity-regularized optimal transport (OT). Though the idea of this extension is straightforward from the earlier DROT work, the authors demonstrated that their GPU implementation can be orders of faster than existing methods for solving the same problem, which shows the superiority of the proposed method.